# FITS: MODELING TIME SERIES WITH $10k$ PARAMETERS

**Zhijian Xu[1], Ailing Zeng[2], Qiang Xu\*[1]**
[1]Department of Computer Science and Engineering, CUHK
[2]International Digital Economy Academy (IDEA)
`zjxu21@cse.cuhk.edu.hk, zengailing@idea.edu.cn, qxu@cse.cuhk.edu.hk`

## ABSTRACT

In this paper, we introduce FITS, a lightweight yet powerful model for time series analysis. Unlike existing models that directly process raw time-domain data, FITS operates on the principle that time series can be manipulated through interpolation in the complex frequency domain, achieving performance comparable to state-of-the-art models for time series forecasting and anomaly detection tasks. Notably, FITS accomplishes this with a svelte profile of just about $10k$ parameters, making it ideally suited for edge devices and paving the way for a wide range of applications. The code is available at: `https://github.com/VEWOXIC/FITS`.

## 1 INTRODUCTION

Time series analysis plays a pivotal role in a myriad of sectors, from healthcare appliances to smart factories. Within these domains, the reliance is often on edge devices like smart sensors, driven by MCUs with limited computational and memory resources. Time series data, marked by its inherent complexity and dynamism, typically presents information that is both sparse and scattered within the time domain. To effectively harness this data, recent research has given rise to sophisticated models and methodologies (Zhou et al., 2021; Liu et al., 2022a; Zeng et al., 2023; Nie et al., 2023; Zhang et al., 2022). Yet, the computational and memory costs of these models makes them unsuitable for resource-constrained edge devices.

On the other hand, the frequency domain representation of time series data promises a more compact and efficient portrayal of inherent patterns. While existing research has indeed tapped into the frequency domain for time series analysis — FEDformer (Zhou et al., 2022a) enriches its features using spectral data, and TimesNet (Wu et al., 2023) harnesses high-amplitude frequencies for feature extraction via CNNs — a comprehensive utilization of the frequency domain's compactness remains largely unexplored. Specifically, the ability of the frequency domain to employ complex numbers in capturing both amplitude and phase information is not utilized, resulting in the continued reliance on compute-intensive models for temporal feature extraction.

In this study, *we reinterpret time series analysis tasks, such as forecasting and reconstruction, as interpolation exercises within the complex frequency domain*. Essentially, we produce an extended time series segment by interpolating the frequency representation of a provided segment. Specifically, for forecasting, we can obtain the forecasting results by simply extending the given look-back window with frequency interpolation; for reconstruction, we recover the original segment by interpolating the frequency representation of its downsampled counterpart. Building on this insight, we introduce **FITS** (**F**requency **I**nterpolation **T**ime **S**eries Analysis Baseline). The core of FITS is a complex-valued linear layer, meticulously designed to learn amplitude scaling and phase shift, thereby facilitating interpolation within the complex frequency domain.

Notably, while FITS operates interpolations in the frequency domain, it fundamentally remains a time domain model, integrating the rFFT (Brigham & Morrow, 1967) operation. That is, we transform the input segment into the complex frequency domain using rFFT for frequency interpolation. This interpolated frequency data is then mapped back to the time domain, resulting in an elongated segment ready for supervision. This innovative design allows FITS to be highly adaptable, fitting seamlessly into a plethora of downstream time domain tasks such as forecasting and anomaly detection.

Apart from its streamlined linear architecture, FITS incorporates a low-pass filter. This ensures a compact representation while preserving essential information. Despite its simplicity, FITS consistently achieves state-of-the-art (SOTA) performance. Remarkably, in most scenarios, FITS achieves this feat with fewer than **10k parameters**. This makes it **50 times more compact** than the lightweight temporal linear model DLinear (Zeng et al., 2023) and approximately **10,000 times smaller** than other mainstream models. Given its efficiency in memory and computation, FITS stands out as an ideal candidate for deployment, or even for training directly on edge devices, be it for forecasting or anomaly detection.

In summary, our contributions can be delineated as follows:

- We present FITS, an exceptionally lightweight model for time series analysis, boasting a modest parameter count in the range of **5k∼10k**.
- FITS offers a pioneering approach to time series analysis by employing a complex-valued neural network. This simultaneously captures both amplitude and phase information, paving the way for a more comprehensive and efficient representation of time series data.
- Despite being orders of magnitude smaller than most mainstream models, FITS consistently delivers top-tier performance across a range of time series analysis tasks.

## 2 RELATED WORK AND MOTIVATION

### 2.1 FREQUENCY-AWARE TIME SERIES ANALYSIS MODELS

Recent advancements in time series analysis have witnessed the utilization of frequency domain information to capture and interpret underlying patterns. FNet (Lee-Thorp et al., 2022) leverages a pure attention-based architecture to efficiently capture temporal dependencies and patterns solely in the frequency domain, eliminating the need for convolutional or recurrent layers. On the other hand, FEDFormer (Zhou et al., 2022a) and FiLM (Zhou et al., 2022b) incorporate frequency information as supplementary features to enhance the model's capability in capturing long-term periodic patterns and speed up computation.

The other line of work aims to capture the periodicity inherent in the data. For instance, DLinear (Zeng et al., 2023) adopts a single linear layer to extract the dominant periodicity from the temporal domain and surpasses a range of deep feature extraction-based methods. More recently, TimesNet (Wu et al., 2023) achieves state-of-the-art results by identifying several dominant frequencies instead of relying on a single dominant periodicity. Specifically, they use the Fast Fourier Transform (FFT) to find the frequencies with the largest energy and reshape the original 1D time series into 2D images according to their periods.

However, these approaches still rely on feature engineering to identify the dominant period set. Selecting this set based on energy may only consider the dominant period and its harmonics, limiting the information captured. Moreover, these methodologies are still considered inefficient and prone to overfitting.

### 2.2 DIVIDE AND CONQUER THE FREQUENCY COMPONENTS

Treating a time series as a signal allows us to break it down into a linear combination of sinusoidal components without any information loss. Each component possesses a unique frequency, initial phase, and amplitude. Forecasting directly on the original time series can be challenging, but forecasting each frequency component is comparatively straightforward, as we only need to apply a phase bias to the sinusoidal wave based on the time shift. Subsequently, we linearly combine these shifted sinusoidal waves to obtain the forecasting result.

This approach effectively preserves the frequency characteristics of the given look-back window while maintaining semantic consistency between the look-back window and the forecasting horizon. Specifically, the resulting forecasted values maintain the frequency features of the original time series with a reasonable time shift, ensuring that semantic consistency is maintained.

However, forecasting each sinusoidal component in the time domain can be cumbersome, as the sinusoidal components are treated as a sequence of data points. To address this, we propose conducting

this manipulation in the complex frequency domain, which offers a more compact and information-rich representation, as described below.

## 3 METHOD

### 3.1 PRELIMINARY: FFT AND COMPLEX FREQUENCY DOMAIN

The Fast Fourier Transform (FFT, (Brigham & Morrow, 1967)) efficiently computes the Discrete Fourier Transform (DFT) of complex number sequences. The DFT transforms discrete-time signals from the time domain to the complex frequency domain. In time series analysis, the Real FFT (rFFT) is often employed when working with real input signals. It condenses an input of N real numbers into a sequence of N/2+1 complex numbers, representing the signal in the complex frequency domain.

**Complex Frequency Domain**

In Fourier analysis, the complex frequency domain is a representation of a signal in which each frequency component is characterized by a complex number. This complex number captures both the amplitude and phase of the component, providing a comprehensive description. The amplitude of a frequency component represents the magnitude or strength of that component in the original time-domain signal. In contrast, the phase represents the temporal shift or delay introduced by that component. Mathematically, the complex number associated with a frequency component can be represented as a complex exponential element with a given amplitude and phase:

$$X(f) = |X(f)|e^{j\theta(f)},$$

where $X(f)$ is the complex number associated with the frequency component at frequency $f$, $|X(f)|$ is the amplitude of the component, and $\theta(f)$ is the phase of the component. As shown in Fig. 1(a), in the complex plane, the complex exponential element can be visualized as a vector with a length equal to the amplitude and angle equal to the phase:

$$X(f) = |X(f)|(\cos\theta(f) + j\sin\theta(f))$$

Therefore, the complex number in the complex frequency domain provides a concise and elegant means of representing the amplitude and phase of each frequency component in the Fourier transform.

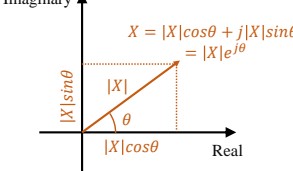
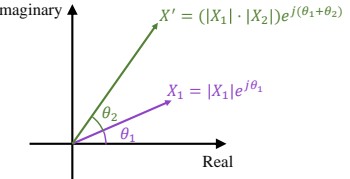

(a) Complex number on the complex plane

(b) Complex number multiplication

Figure 1: Illustration of Complex Number Visualization and Multiplication

**Time Shift and Phase Shift**. The time shift of a signal corresponds to the phase shift in the frequency domain. Especially in the complex frequency domain, we can express such phase shift by multiplying a unit complex exponential element with the corresponding phase. Mathematically, if we shift a signal $x(t)$ forward in time by a constant amount $\tau$, resulting in the signal $x(t - \tau)$, the Fourier transform is given by:

$$X_\tau(f) = e^{-j2\pi f\tau}X(f) = |X(f)|e^{j(\theta(f)-2\pi f\tau)} = [cos(-2\pi f\tau) + jsin(-2\pi f\tau)]X(f)$$

The shifted signal still has an amplitude of $|X(f)|$, while the phase $\theta_\tau(f) = \theta(f) - 2\pi f\tau$ shows a shift which is linear to the time shift.

In summary, the amplitude scaling and phase shifting can be simultaneously expressed as the multiplication of complex numbers, as shown in Fig. 1(b).

## 3.2 FITS PIPELINE

Motivated by the fact that a longer time series provides a higher frequency resolution in its frequency representation, we train FITS to extend time series segment by interpolating the frequency representation of the input time series segment. We use a single layer of complex-valued linear layer to learn such interpolation, so that it can learn amplitude scaling and phase shifting as the multiplication of complex numbers during the interpolation process. As shown in Fig. 2, we use rFFT to project time series segments to the complex frequency domain. After the interpolation, the frequency representation is projected back with inverse rFFT (irFFT).

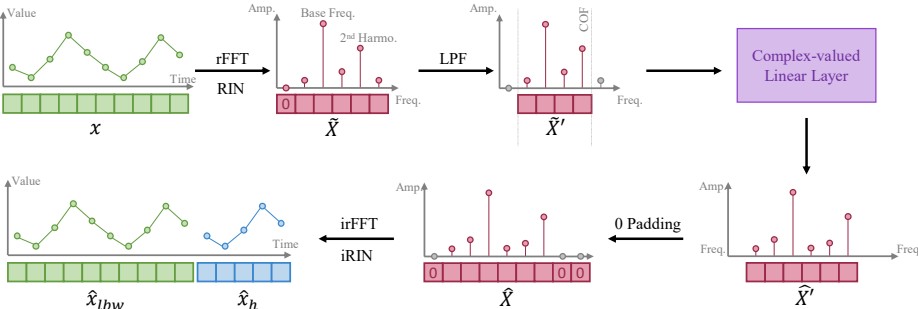

Figure 2: Pipeline of FITS, with a focus on the forecasting task. Initially, the time series is normalized to zero-mean, followed by rFFT for frequency domain projection. After LPF, a single complex-valued linear layer interpolates the frequency. Zero padding and irFFT then revert this back to the time domain, with iRIN finally reversing the normalization. The reconstruction task follows the same pipeline, except for the reconstruction supervision loss. Please check appendix for detail.

However, the mean of such segments will result in a very large 0-frequency component in its complex frequency representation. To address this, we pass it through reversible instance-wise normalization (RIN) (Kim et al., 2022) to obtain a zero-mean instance. As a result, the normalized complex frequency representation now has a length of $N/2$, where $N$ represents the original length of the time series.

Additionally, FITS integrates a low-pass filter (LPF) to further reduce its model size. The LPF effectively eliminates high-frequency components above a specified cutoff frequency, compacting the model representation while preserving essential time series information. Despite operating in the frequency domain, FITS is supervised in the time domain using standard loss functions like Mean Squared Error (MSE) after the inverse real-to-complex Fast Fourier Transform (irFFT). This allows for versatile supervision tailored to various downstream time series tasks.

In the case of forecasting tasks, we generate the look-back window along with the horizon as shown in Fig. 2. This allows us to provide supervision for forecasting and backcasting, where the model is encouraged to accurately reconstruct the look-back window. Our ablation study reveals that combining backcast and forecast supervision can yield improved performance in certain scenarios.

For reconstruction tasks, we downsample the original time series segment based on a specific downsampling rate. Subsequently, FITS is employed to perform frequency interpolation, enabling the reconstruction of the downsampled segment back to its original form. Thus, direct supervision is applied using reconstruction loss to ensure faithful reconstruction. The reconstruction tasks also follow the pipeline in Fig. 2 with the supervision replaced with reconstruction loss.

## 3.3 KEY MECHANISMS OF FITS

**Complex Frequency Linear Interpolation.** To control the output length of the model, we introduce an interpolation rate denoted as $\eta$, which represents the ratio of the model's output length $L_o$ to its corresponding input length $L_i$. Frequency interpolation operates on the normalized complex frequency representation, which has half the length of the original time series. Importantly, this interpolation rate can also be applied to the frequency domain, as indicated by the equation:

$$\eta_{freq} = \frac{L_o/2}{L_i/2} = \frac{L_o}{L_i} = \eta$$

Based on this formula, with an arbitrary frequency $f$, the frequency band $1 \sim f$ in the original signal is linearly projected to the frequency band $1 \sim \eta f$ in the output signal. As a result, we define the input length of our complex-valued linear layer as $L$ and the interpolated output length as $\eta L$. Notably, when applying the Low Pass Filter (LPF), the value of $L$ corresponds to the cutoff frequency (COF) of the LPF. After performing frequency interpolation, the complex frequency representation is zero-padded to a length of $L_o/2$, where $L_o$ represents the desired output length. Prior to applying the irFFT, an additional zero is introduced as the representation's zero-frequency component.

**Low Pass Filter (LPF).** The primary objective of incorporating the LPF within FITS is to compress the model's volume while preserving essential information. The LPF achieves this by discarding frequency components above a specified cutoff frequency (COF), resulting in a more concise frequency domain representation. The LPF retains the relevant information in the time series while discarding components beyond the model's learning capability. This ensures that a significant portion of the original time series' meaningful content is preserved. As demonstrated in Fig. 3, the filtered waveform exhibits minimal distortion even when only preserving a quarter of the original frequency domain representation. Furthermore, the high-frequency components filtered out by the LPF typically comprise noise, which are inherently irrelevant for effective time series modeling.

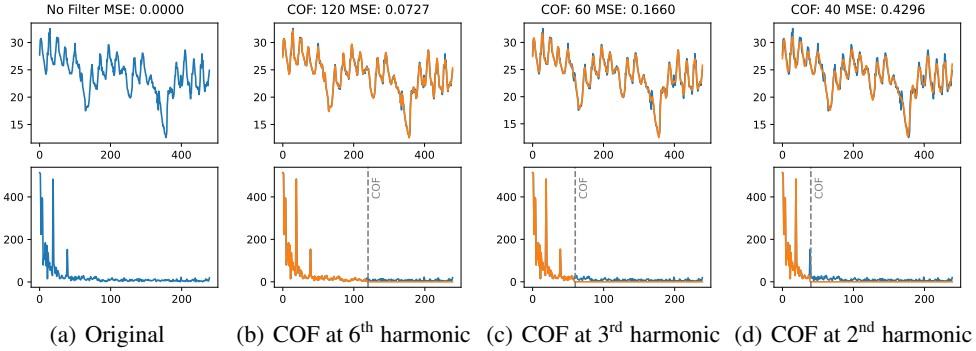

(a) Original    (b) COF at 6th harmonic    (c) COF at 3rd harmonic    (d) COF at 2nd harmonic

Figure 3: Waveform (1st row) and amplitude spectrum (2nd row) of a time series segment selected from the 'OT' channel of the ETTh1 dataset, spanning from the 1500th to the 1980th data point. The segment has a length of 480, and its dominant periodicity is 24, corresponding to a base frequency of 20. The blue lines represent the waveform/spectrum with no applied filter, while the orange lines represent the waveform/spectrum with the filter applied. The filter cutoff frequency is chosen based on a harmonic of the original time series.

Selecting an appropriate cutoff frequency (COF) remains a nontrivial challenge. To address this, we propose a method based on the harmonic content of the dominant frequency. Harmonics, which are integer multiples of the dominant frequency, play a significant role in shaping the waveform of a time series. By aligning the cutoff frequency with these harmonics, we keep relevant frequency components associated with the signal's structure and periodicity. This approach leverages the inherent relationship between frequencies to extract meaningful information while suppressing noise and irrelevant high-frequency components. The impact of COF on different harmonics' waveforms is shown in Fig. 3. We further elaborate on the impact of COF in our experimental results.

**Weight Sharing.** FITS handles multivariate tasks by sharing weights as in (Zeng et al., 2023), balancing performance and efficiency. In practice, channels often share a common base frequency when originating from the same physical system, such as 50/60Hz for electrical appliances or daily base frequencies for city traffic. Most of the datasets used in our experiments belong to this category. For datasets that indeed contain channels with different base frequencies, we can cluster those channels according to the base frequency and train an individual FITS model for each cluster.

## 4 Experiments for Forecasting

### 4.1 Forecasting as Frequency Interpolation

Typically, the forecasting horizon is shorter than the given look-back window, rendering direct interpolation unsuitable. Instead, we formulate the forecasting task as the interpolation of a look-back

window, with length $L$, to a combination of the look-back window and forecasting horizon, with length $L + H$. This design enables us to provide more supervision during training. With this approach, we can supervise not only the forecasting horizon but also the backcast task on the look-back window. Our experimental results demonstrate that this unique training strategy contributes to the improved performance of FITS. The interpolation rate of the forecasting task is calculated by:

$$\eta_{Fore} = 1 + \frac{H}{L},$$

where $L$ represents the length of the look-back window and $H$ represents the length of the forecasting horizon.

## 4.2 EXPERIMENT SETTINGS

**Datasets.** All datasets used in our experiments are widely-used and publicly available real-world datasets, including, Traffic, Electricity, Weather, ETT (Zhou et al., 2021). We summarize the characteristics of these datasets in appendix. Apart from these datasets for long-term time series forecasting, we also use the M4 dataset to test the short-term forecasting performance.

**Baselines**. To evaluate the performance of FITS in comparison to state-of-the-art time series forecasting models, including PatchTST (Nie et al., 2023), TimesNet (Wu et al., 2023), FEDFormer (Zhou et al., 2022a) and LTSF-Linear (Zeng et al., 2023), we rerun all the experiment with code and scripts provided by their official implementation [1]. We report the comparison with NBeats (Oreshkin et al., 2019), NHits (Challu et al., 2023) and other transformer-based methods in the appendix.

**Evaluation metrics**. We follow the previous works (Zhou et al., 2022a; Zeng et al., 2023; Zhang et al., 2022) to compare forecasting performance using Mean Squared Error (MSE) as the core metrics. Moreover, to evaluate the short-term forecasting, we symmetric Mean Absolute Percentage Error (SMAPE) following TimesNet (Wu et al., 2023).

**Implementation details**. We conduct grid search on the look-back window of 90, 180, 360, 720 and cutoff frequency, the only hyper-parameter. Further experiments also show that a longer look-back window can result in better performance in most cases. To avoid information leakage, We choose the hyper-parameter based on the performance of the validation set. We report the result of FITS as the mean and standard deviation of 5 runs with random chosen random seeds.

## 4.3 COMPARISONS WITH SOTAS

**Competitive Performance with High Efficiency**

We present the results of our experiments on long-term forecasting in Tab. 1 and Tab. 2. The results for short-term forecasting on the M4 dataset are provided in the Appendix. Remarkably, our FITS consistently achieves comparable or even superior performance across all experiments.

Tab. 3 presents the number of trainable parameters and MACs [2] for various TSF models using a look-back window of 96 and a forecasting horizon of 720 on the Electricity dataset. The table clearly demonstrates the exceptional efficiency of FITS compared to other models.

Among the listed models, the parameter counts range from millions down to thousands. Notably, large models such as TimesNet and Pyraformer require a staggering number of parameters, with 300.6M and 241.4M, respectively. Similarly, popular models like Informer, Autoformer, and FEDformer have parameter counts in the range of 13.61M to 20.68M. Even the lightweight yet state-of-the-art model PatchTST has a parameter count of over 1 million.

In contrast, FITS stands out as a highly efficient model with an impressively low parameter count. With only 4.5K to 16K parameters, FITS achieves comparable or even superior performance compared to these larger models. It is worth highlighting that FITS requires significantly fewer parameters compared to the next smallest model, Dlinear, which has 139.7K parameters. For instance, when considering a 720 look-back window and a 720 forecasting horizon, the Dlinear model requires over 1 million parameters, whereas FITS achieves similar performance with only 10k-50k parameters.

---

[1]With a long-standing bug in the coding architecture fixed, see README file in our codebase.

[2]MACs (Multiply-Accumulate Operations) is a commonly used metric that counts the total number of multiplication and addition operations in a neural network.

Table 1: Long-term forecasting results on ETT dataset in MSE. The best result is highlighted in **bold**, and the second best is highlighted with underline. IMP is the improvement between FITS and the second best/ best result, where a larger value indicates a better improvement. Most of the STD are under 5e-4 and shown as 0.000 in this table.

| Dataset | ETTh1 | | | | ETTh2 | | | | ETTm1 | | | | ETTm2 | | | |
|---|---|---|---|---|---|---|---|---|---|---|---|---|---|---|---|---|
| Horizon | 96 | 192 | 336 | 720 | 96 | 192 | 336 | 720 | 96 | 192 | 336 | 720 | 96 | 192 | 336 | 720 |
| PatchTST | 0.385 | 0.413 | 0.440 | 0.456 | 0.274 | 0.338 | 0.367 | 0.391 | **0.292** | **0.330** | **0.365** | 0.419 | 0.163 | 0.219 | 0.276 | 0.368 |
| Dlinear | 0.384 | 0.443 | 0.446 | 0.504 | 0.282 | 0.350 | 0.414 | 0.588 | 0.301 | 0.335 | 0.371 | 0.426 | 0.171 | 0.237 | 0.294 | 0.426 |
| FedFormer | 0.375 | 0.427 | 0.459 | 0.484 | 0.340 | 0.433 | 0.508 | 0.480 | 0.362 | 0.393 | 0.442 | 0.483 | 0.189 | 0.256 | 0.326 | 0.437 |
| TimesNet | 0.384 | 0.436 | 0.491 | 0.521 | 0.340 | 0.402 | 0.452 | 0.462 | 0.338 | 0.374 | 0.410 | 0.478 | 0.187 | 0.249 | 0.321 | 0.408 |
| FITS | **0.372** | **0.404** | **0.427** | **0.424** | **0.271** | **0.331** | **0.354** | **0.377** | 0.303 | 0.337 | 0.366 | **0.415** | **0.162** | **0.216** | **0.268** | **0.348** |
| STD | 0.000 | 0.000 | 0.000 | 0.000 | 0.000 | 0.000 | 0.000 | 0.000 | 0.000 | 0.000 | 0.000 | 0.000 | 0.000 | 0.000 | 0.000 | 0.000 |
| IMP | 0.003 | 0.009 | 0.013 | 0.032 | 0.003 | 0.007 | 0.013 | 0.014 | -0.011 | -0.007 | -0.001 | 0.004 | 0.001 | 0.003 | 0.008 | 0.020 |

Table 2: Long-term forecasting results on three popular datasets in MSE. The best result is highlighted in **bold** and the second best is highlighted with underline. IMP is the improvement between FITS and the second best/ best result, where a larger value indicates a better improvement. Most of the STD are under 5e-4 and shown as 0.000 in this table.

| Dataset | Weather | | | | Electricity | | | | Traffic | | | |
|---|---|---|---|---|---|---|---|---|---|---|---|---|
| Horizon | 96 | 192 | 336 | 720 | 96 | 192 | 336 | 720 | 96 | 192 | 336 | 720 |
| PatchTST | 0.151 | 0.195 | 0.249 | 0.321 | **0.129** | **0.149** | 0.166 | 0.210 | **0.366** | **0.388** | **0.398** | 0.457 |
| Dlinear | 0.174 | 0.217 | 0.262 | 0.332 | 0.140 | 0.153 | 0.169 | 0.204 | 0.413 | 0.423 | 0.437 | 0.466 |
| Fedformer | 0.246 | 0.292 | 0.378 | 0.447 | 0.188 | 0.197 | 0.212 | 0.244 | 0.573 | 0.611 | 0.621 | 0.630 |
| TimesNet | 0.172 | 0.219 | 0.280 | 0.365 | 0.168 | 0.184 | 0.198 | 0.220 | 0.593 | 0.617 | 0.629 | 0.640 |
| FITS | **0.143** | **0.186** | **0.236** | **0.307** | 0.134 | **0.149** | **0.165** | **0.203** | 0.385 | 0.397 | 0.410 | **0.448** |
| STD | 0.001 | 0.001 | 0.001 | 0.001 | 0.000 | 0.000 | 0.000 | 0.000 | 0.001 | 0.000 | 0.000 | 0.001 |
| IMP | 0.008 | 0.009 | 0.013 | 0.014 | -0.005 | 0.000 | 0.001 | 0.001 | -0.019 | -0.009 | -0.012 | 0.009 |

This analysis showcases the remarkable efficiency of FITS. Despite its small size, FITS consistently achieves competitive results, making it an attractive option for time series analysis tasks. FITS demonstrates that achieving state-of-the-art or close to state-of-the-art performance with a considerably reduced parameter footprint is possible, making it an ideal choice for resource-constrained environments.

**Case Study on ETTh2 Dataset**

We conduct a comprehensive case study on the performance of FITS using the ETTh2 dataset, which further highlights the impact of the look-back window and cutoff frequency on model performance. We provide a case study on other datasets in the Appendix. In our experiments, we observe that increasing the look-back window generally leads to improved performance, while the effect of increasing the cutoff frequency is minor.

Table 3: Number of trainable parameters, MACs, and inference time of TSF models under look-back window=96 and forecasting horizon=720 on the Electricity dataset.

| Model | Parameters | MACs | Infer. Time |
|---|---|---|---|
| TimesNet | 301.7M | 1226.49G | N/A |
| Pyraformer | 241.4M | 0.80G | 3.4ms |
| Informer | 14.38M | 3.93G | 49.3ms |
| Autoformer | 14.91M | 4.41G | 164.1ms |
| FiLM | 14.91M | 5.97G | 123.0ms |
| FEDformer | 20.68M | 4.41G | 40.5ms |
| PatchTST | 1.5M | 5.07G | 3.3ms |
| DLinear | 139.7K | 40M | *0.4ms* (3.05ms CPU) |
| FITS (Ours) | **4.5K∼10K** | **1.6M∼8.9M** | 0.6ms (*2.55ms* CPU) |

Tab. 4 showcases the performance results obtained with different look-back window sizes and cutoff frequencies. Larger look-back windows tend to yield better performance across the board. On the other hand, increasing the cutoff frequency only results in marginal performance improvements. However, it is important to note that higher cutoff frequencies come at the expense of increased computational resources, as illustrated in Tab. 5.

Considering these observations, we find utilizing a longer look-back window in combination with a low cutoff frequency to achieve near state-of-the-art performance with minimal computational cost. For instance, FITS surpasses other methods when employing a 720 look-back window and setting the cutoff frequency to the second harmonic. Remarkably, FITS achieves state-of-the-art performance with a parameter count of only around 10k. Moreover, by reducing the look-back window to 360, FITS already achieves close-to-state-of-the-art performance by setting the cutoff frequency to the second harmonic, resulting in a further reduction of the model's parameter count to under 5k (as shown in Tab. 5).

Table 4: The results on the ETTh2 dataset. Values are visualized with a green background, where darker background indicates worse performance. The top-5 best results are highlighted with a red background, and the absolute best result is highlighted with **red bold** font. **F** represents supervision on the forecasting task, while **B+F** represents supervision on backcasting and forecasting tasks.

| Horizon | Look-back Window | 90 | | 180 | | 360 | | 720 | |
| | COF/nth Harmonic | F | B+F | F | B+F | F | B+F | F | B+F |
|---|---|---|---|---|---|---|---|---|---|
| 96 | 2 | 0.293889 | 0.291371 | 0.290314 | 0.288107 | 0.279141 | 0.276635 | 0.275600 | 0.274817 |
| | 3 | 0.293242 | 0.291333 | 0.289803 | 0.287171 | 0.278128 | 0.275723 | 0.273972 | 0.273567 |
| | 4 | 0.292438 | 0.290559 | 0.288541 | 0.286174 | 0.277293 | 0.274494 | 0.272384 | 0.272031 |
| | 5 | 0.292387 | 0.290369 | 0.288530 | 0.285527 | 0.276594 | 0.274042 | 0.272085 | 0.271719 |
| | 6 | 0.292517 | 0.290466 | 0.287814 | 0.285384 | 0.275930 | 0.273883 | 0.271312 | **0.271028** |
| 192 | 2 | 0.379401 | 0.377047 | 0.361995 | 0.359322 | 0.337767 | 0.336419 | 0.334493 | 0.334621 |
| | 3 | 0.379080 | 0.376874 | 0.360790 | 0.358059 | 0.337391 | 0.335736 | 0.333573 | 0.333758 |
| | 4 | 0.378816 | 0.376472 | 0.360524 | 0.357973 | 0.336085 | 0.334531 | 0.332310 | 0.332475 |
| | 5 | 0.378529 | 0.376429 | 0.360234 | 0.357533 | 0.336286 | 0.334475 | 0.332122 | 0.332281 |
| | 6 | 0.378581 | 0.376481 | 0.360049 | 0.357478 | 0.335526 | 0.333846 | **0.331421** | 0.331667 |
| 336 | 2 | 0.419131 | 0.417096 | 0.391167 | 0.388905 | 0.360300 | 0.359665 | 0.356390 | 0.356319 |
| | 3 | 0.419264 | 0.416645 | 0.389740 | 0.387614 | 0.359802 | 0.359291 | 0.355825 | 0.355972 |
| | 4 | 0.419237 | 0.416085 | 0.389790 | 0.387815 | 0.358774 | 0.358096 | 0.354695 | 0.354880 |
| | 5 | 0.418985 | 0.416009 | 0.388972 | 0.387115 | 0.358652 | 0.358093 | 0.354805 | 0.354794 |
| | 6 | 0.418359 | 0.416369 | 0.388943 | 0.387183 | 0.358011 | 0.357432 | **0.354055** | 0.354205 |
| 720 | 2 | 0.420888 | 0.418226 | 0.405711 | 0.404412 | 0.387592 | 0.386235 | 0.379710 | 0.380367 |
| | 3 | 0.420441 | 0.418290 | 0.404405 | 0.403520 | 0.386570 | 0.385907 | 0.379501 | 0.380132 |
| | 4 | 0.420404 | 0.417756 | 0.404631 | 0.403425 | 0.386556 | 0.384828 | 0.378209 | 0.378890 |
| | 5 | 0.419888 | 0.417725 | 0.403562 | 0.402755 | 0.385489 | 0.384758 | 0.378227 | 0.378810 |
| | 6 | 0.419376 | 0.417854 | 0.403643 | 0.402616 | 0.384709 | 0.383960 | **0.377463** | 0.378101 |

These results emphasize the lightweight nature of FITS, making it highly suitable for deployment and training on edge devices with limited computational resources. By carefully selecting the look-back window and cutoff frequency, FITS can achieve excellent performance while maintaining computational efficiency, making it an appealing choice for real-world applications.

# 5 EXPERIMENT
## FOR ANOMALY DETECTION

### 5.1 RECONSTRUCTION
### AS FREQUENCY INTERPOLATION

As discussed before, we tackle the anomaly detection tasks in the self-supervised reconstructing approach. Specifically, we make a $N$ time equidistant sampling on the input and train a FITS network with an interpolation rate of $\eta_{Rec} = N$ to up-sample it. Please check appendix A for detail.

Table 5: The number of parameters under different settings on ETTh1 & ETTh2 dataset.

| Horizon | COF/nth Harmonic | Look-back Window | | | |
| | | 90 | 180 | 360 | 720 |
|---|---|---|---|---|---|
| 96 | 2 | 703 | 1053 | 2279 | 5913 |
| | 3 | 1035 | 1820 | 4307 | 12064 |
| | 4 | 1431 | 2752 | 6975 | 20385 |
| | 5 | 1922 | 3876 | 10374 | 31042 |
| | 6 | 2450 | 5192 | 14338 | 43734 |
| 192 | 2 | 1064 | 1431 | 2752 | 6643 |
| | 3 | 1564 | 2450 | 5192 | 13520 |
| | 4 | 2187 | 3698 | 8475 | 22815 |
| | 5 | 2914 | 5253 | 12558 | 34694 |
| | 6 | 3710 | 7021 | 17334 | 48856 |
| 336 | 2 | 1615 | 1998 | 3483 | 7665 |
| | 3 | 2392 | 3395 | 6608 | 15704 |
| | 4 | 3321 | 5160 | 10725 | 26460 |
| | 5 | 4402 | 7293 | 15834 | 40006 |
| | 6 | 5600 | 9794 | 21828 | 56539 |
| 720 | 2 | 3078 | 3510 | 5418 | 10512 |
| | 3 | 4554 | 5950 | 10266 | 21424 |
| | 4 | 6318 | 9030 | 16650 | 36180 |
| | 5 | 8370 | 12750 | 24570 | 54780 |
| | 6 | 10710 | 17110 | 34026 | 77224 |

### 5.2 EXPERIMENT SETTINGS

**Datasets**. We use five commonly used benchmark datasets: SMD (Server Machine Dataset (Su et al., 2019)), PSM (Polled Server Metrics (Abdulaal et al., 2021)), SWaT (Secure Water Treatment (Mathur & Tippenhauer, 2016)), MSL (Mars Science Laboratory rover), and SMAP (Soil Moisture Active Passive satellite) (Hundman et al., 2018). We report the performance on the synthetic dataset (Lai et al., 2021) in the appendix G.

**Baselines**. We compare FITS with models such as TimesNet (Wu et al., 2023), Anomaly Transformer (Xu et al., 2022), THOC (Shen et al., 2020), Omnianomaly (Su et al., 2019), DGHL (Challu et al., 2022). Following TimesNet (Wu et al., 2023), we also compare the anomaly detection performance with other models (Zeng et al., 2023; Zhang et al., 2022; Woo et al., 2022; Zhou et al., 2022a).

**Evaluation metrics**. Following the previous works (Xu et al., 2022; Shen et al., 2020; Wu et al., 2023), we use Precision, Recall, and F1-score as metrics.

**Implementation details**. We use a window size of 200 and downsample the time series segment by a factor of 4 as the input to train FITS to reconstruct the original segment. We follow the methodology of the Anomaly Transformer (Xu et al., 2022), where time points exceeding a certain reconstruction loss threshold are classified as anomalies. The threshold is selected based on the highest F1 score achieved on the validation set. To handle consecutive abnormal segments, we adopt a widely-used adjustment strategy (Su et al., 2019; Xu et al., 2018; Shen et al., 2020), considering all anomalies within a specific successive abnormal segment as correctly detected when one anomalous time point is identified. This approach aligns with real-world applications, where an abnormal time point often triggers the attention to the entire segment.

Table 6: Anomaly detection result of F1-scores on 5 datasets. The best result is highlighted in **bold**, and the second best is highlighted with underline. Full results are reported in the Appendix.

| Models | FITS | TimesNet | Anomaly Transformer | THOC | Omni Anomaly | Stationary Transformer | DGHL | OCSVM | IForest | LightTS | Dlinear | IMP |
|---|---|---|---|---|---|---|---|---|---|---|---|---|
| SMD | **99.95** | 85.81 | 92.33 | 84.99 | 85.22 | 84.72 | N/A | 56.19 | 53.64 | 82.53 | 77.1 | 7.62 |
| PSM | 93.96 | 97.47 | 97.89 | **98.54** | 80.83 | 97.29 | N/A | 70.67 | 83.48 | 97.15 | 93.55 | -3.93 |
| SWaT | **98.9** | 91.74 | 94.07 | 85.13 | 82.83 | 79.88 | 87.47 | 47.23 | 47.02 | 93.33 | 87.52 | 4.83 |
| SMAP | 70.74 | 71.52 | **96.69** | 90.68 | 86.92 | 71.09 | 96.38 | 56.34 | 55.53 | 69.21 | 69.26 | -25.95 |
| MSL | 78.12 | 85.15 | 93.59 | 89.69 | 87.67 | 77.5 | **94.08** | 70.82 | 66.45 | 78.95 | 84.88 | -15.96 |

## 5.3 COMPARISONS WITH SOTAS

In Table 6, FITS stands out with outstanding results on various datasets. Particularly, on SMD and SWaT datasets, FITS achieves nearly perfect F1-scores, around 99.95% and 98.9%, respectively, showcasing its precision in anomaly detection and classification. In contrast, models like TimesNet, Anomaly Transformer, and Stationary Transformer struggle to match FITS' performance on these datasets.

However, FITS shows comparatively lower performance on the SMAP and MSL datasets. These datasets present a challenge due to their binary event data nature, which may not be effectively captured by FITS' frequency domain representation. In such cases, time-domain modeling is preferable as the raw data format is sufficiently compact. Thus, models specifically designed for anomaly detection, such as THOC and Omni Anomaly, achieve higher F1-scores on these datasets.

For a more comprehensive evaluation, waveform visualizations and detailed analysis can be found in the appendix, providing deeper insights into FITS' strengths and limitations in different anomaly detection scenarios. It is important to note that the reported results are achieved with a parameter range of 1-4K and MACs (Multiply-Accumulate Operations) of 10-137K, which will be further detailed in the appendix.

While the datasets in use are instrumental, it is imperative to acknowledge their limitations as delineated in (Lai et al., 2021). Particularly on the synthetic dataset from (Lai et al., 2021), FITS demonstrates impeccable detection capabilities, registering a flawless 100% F1 score. For a detailed breakdown, readers can refer to the table in appendix G. This dataset marries a sinusoidal wave of a single frequency with intricately introduced anomaly patterns, which pose challenges for identification in the time domain. Yet, FITS, leveraging the frequency domain, adeptly discerns these anomalies, particularly those introducing unexpected frequency components.

Moreover, FITS boasts an impressive sub-millisecond inference speed — a marked distinction when compared to the latency typical of larger models or communication overheads. This speed underscores FITS's suitability as a first-responder tool for promptly spotting critical errors. When paired as a preliminary filter with a specialized AD algorithm geared for detailed detection, the combined system stands as a paragon of both robustness and swift responsiveness facing diverse anomalies.

## 6 CONCLUSIONS AND FUTURE WORK

In this paper, we propose FITS for time series analysis, a low-cost model with $10k$ parameters that can achieve performance comparable to state-of-the-art models that are often several orders of magnitude larger. As the future work, we plan to evaluate FITS on more real-world scenario and improve the interpretability of it. Further, we also aim to explore the frequency domain large-scale complex-valued neural network such as complex-valued Transformers.

ACKNOWLEDGEMENTS

This work is supported in part by the CUHK SSFCRS funding under Grant No. 3136023, and in part by the Research Matching Grant Scheme under Grant No. 7106937, 8601130, and 8601440.

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

## A   PIPELINE FOR RECONSTRUCTION

The pipeline for the reconstruction task is shown in Fig. 4. In this process, the model input $x$ is derived from a segment of the time series $y$ using an equidistant sampling technique with a specified downsample rate $\eta$. Subsequently, FITS performs frequency interpolation, generating an upsampled output $\hat{x}_{up-sampled}$ with the same length as $y$. The reconstruction loss is computed by comparing the original $y$ and the upsampled $\hat{x}_{up-sampled}$. Please note that, due to space constraints, the depicted downsample/upsample rate $\eta$ in the figure is shown as 1.5, which is not a practical value. In our actual experiments, we employ a $\eta$ value of 4.

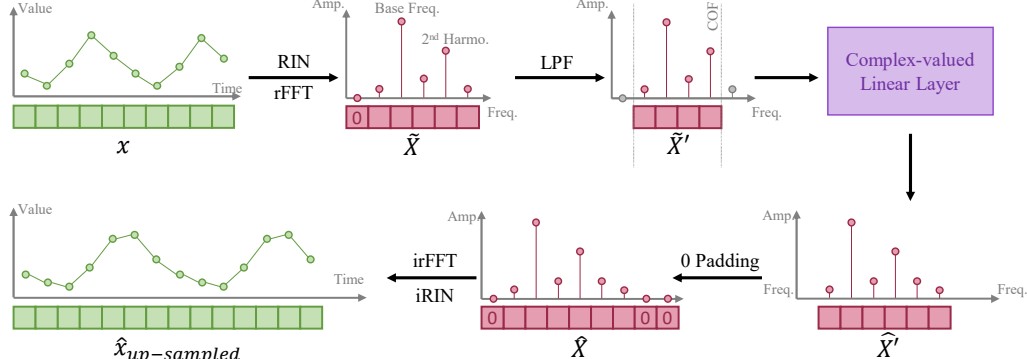

Figure 4: Pipeline of FITS, with a focus on the Reconstruction task.

## B   DETAILS OF FORECASTING DATASETS

We report the characteristics in the tab.7.

Table 7: The statistics of the seven used forecasting datasets.

| Dataset | Traffic | Electricity | Weather | ETTh1&ETTh2 | ETTm1 &ETTm2 |
|---|---|---|---|---|---|
| Channels | 862 | 321 | 21 | 7 | 7 |
| Sampling Rate | 1hour | 1hour | 10min | 1hour | 15min |
| Total Timesteps | 17,544 | 26,304 | 52,696 | 17,420 | 69,680 |

## C   CRITICAL DIFFERENCE PLOT

We generate the critical difference plot on our result with the default alpha as 0.05 as shown in Fig. 5. FITS's placement at the top of the critical difference plot, without intersecting with other lines, demonstrates its consistent and superior performance in terms of MSE compared to the other models. This signifies the effectiveness of FITS in forecasting tasks. Moreover, the absence of intersection indicates the statistical significance of the performance difference, indicating that the disparity in MSE between FITS and others is unlikely due to chance alone. The critical difference plot also showcases the robustness of FITS's performance across various evaluation metrics, reinforcing its reliability. As the top performer in terms of MSE, FITS emerges as a strong contender for model selection when tackling regression problems. The statistical significance illustrated by the critical difference plot further bolsters the confidence in the performance comparison, providing substantial evidence that FITS outperforms the alternatives significantly.

## D   MORE RESULTS ON FORECASTING TASK

We show the comparison with transformer-based models, short-term forecasting on M4, and the impact of random seeds below.

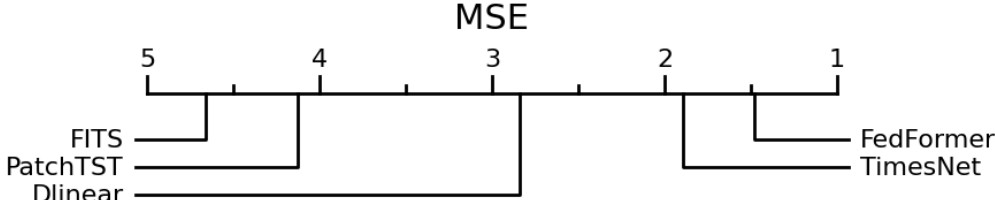

Figure 5: The Critical Difference Plot on the FITS and other baselines with alpha=0.05.

## D.1 COMPARISON WITH TRANSFORMER-BASED METHODS

We further compare FITS with Autoformer (Wu et al., 2021), Informer (Zhou et al., 2021), FiLM (Zhou et al., 2022b) and Pyraformer (Liu et al., 2022b). The results are shown in Tab. 8 and Tab. 9. Note that the results in these tables are directly reterived from the original paper and may still suffer from the bug mentioned above. We cannot rerun these models because of the incomplete codebase or the extereme large time consumption.

Table 8: Long-term forecasting results on ETT datasets in MSE. The best result is highlighted in **bold**.

| Dataset | ETTh1 | | | | ETTh2 | | | | ETTm1 | | | | ETTm2 | | | |
|---|---|---|---|---|---|---|---|---|---|---|---|---|---|---|---|---|
| Horizon | 96 | 192 | 336 | 720 | 96 | 192 | 336 | 720 | 96 | 192 | 336 | 720 | 96 | 192 | 336 | 720 |
| Autoformer | 0.449 | 0.500 | 0.521 | 0.514 | 0.358 | 0.456 | 0.482 | 0.515 | 0.505 | 0.553 | 0.621 | 0.671 | 0.255 | 0.281 | 0.339 | 0.433 |
| Informer | 0.865 | 1.008 | 1.107 | 1.181 | 3.755 | 5.602 | 4.721 | 3.647 | 0.672 | 0.795 | 1.212 | 1.166 | 0.365 | 0.533 | 1.363 | 3.379 |
| FEDFormer | 0.376 | 0.420 | 0.459 | 0.506 | 0.346 | 0.429 | 0.496 | 0.463 | 0.379 | 0.426 | 0.445 | 0.543 | 0.203 | 0.269 | 0.325 | 0.421 |
| Pyraformer | 0.664 | 0.790 | 0.891 | 0.963 | 0.645 | 0.788 | 0.907 | 0.963 | 0.543 | 0.557 | 0.754 | 0.908 | 0.435 | 0.730 | 1.201 | 3.625 |
| FiLM | 0.371 | 0.414 | 0.442 | 0.465 | 0.284 | 0.357 | 0.377 | 0.439 | 0.302 | 0.338 | 0.373 | 0.420 | 0.165 | 0.222 | 0.277 | 0.371 |
| FITS | **0.368** | **0.404** | **0.405** | **0.425** | **0.255** | **0.307** | **0.306** | **0.368** | **0.305** | **0.339** | **0.366** | **0.414** | **0.164** | **0.217** | **0.269** | **0.347** |

Table 9: Long-term forecasting results on three popular datasets in MSE. The best result is highlighted in **bold**.

| Dataset | Electricity | | | | Traffic | | | | Weather | | | |
|---|---|---|---|---|---|---|---|---|---|---|---|---|
| Horizon | 96 | 192 | 336 | 720 | 96 | 192 | 336 | 720 | 96 | 192 | 336 | 720 |
| Autoformer | 0.201 | 0.222 | 0.231 | 0.254 | 0.613 | 0.616 | 0.622 | 0.660 | 0.266 | 0.307 | 0.359 | 0.419 |
| Informer | 0.274 | 0.296 | 0.300 | 0.373 | 0.719 | 0.696 | 0.777 | 0.864 | 0.300 | 0.598 | 0.578 | 1.059 |
| FEDFormer | 0.193 | 0.201 | 0.214 | 0.246 | 0.587 | 0.604 | 0.621 | 0.626 | 0.217 | 0.276 | 0.339 | 0.403 |
| Pyraformer | 0.386 | 0.386 | 0.378 | 0.376 | 2.085 | 0.867 | 0.869 | 0.881 | 0.896 | 0.622 | 0.739 | 1.004 |
| FiLM | 0.154 | 0.164 | 0.188 | 0.236 | 0.416 | 0.408 | 0.425 | 0.520 | 0.199 | 0.228 | 0.267 | 0.319 |
| FITS | **0.137** | **0.142** | **0.165** | **0.202** | **0.381** | **0.381** | **0.410** | **0.446** | **0.145** | **0.188** | **0.236** | **0.308** |

## D.2 COMPARISON WITH NBEATS & NHITS

We show the comparison with mentioned N-HiTS and N-BEATS on MSE in the following table. FITS outperforms these two models in most cases while maintaining a compact model size. We will consider adding the following results to our main result. The results for N-HiTS and N-BEATS are retrieved from the paper of N-HiTS (Challu et al., 2023).

## D.3 SHORT-TERM FORECASTING ON M4

We evaluate FITS' performance on the M4 dataset following the TimesNet (Wu et al., 2023). We retrieve the following results from the TimesNet paper. As shown in Tab.11, FITS shows the suboptimal results on the M4 dataset. The reason for this outcome is threefold. First, the M4 dataset is a collection of many time series from different domains. These time series have different temporal information and periodicity, and no correlations exist among them. We can not regard them as simple multivariate forecasting tasks. Second, other models have a very large amount of parameters, especially TimesNet, which makes them have enough capability to model such diverse datasets with one model. However, considering the lightweight of FITS, it is hard for it to achieve ideal results. Finally, the setting for the M4 dataset is not suitable for FITS. The look-back window is set to

Table 10: Comparison with N-HiTS and N-BEATS on MSE

| Dataset | Horizon | FITS | N-BEATS | N-HiTS |
|---------|---------|------|---------|--------|
| Electricity | 96 | **0.138** | 0.145 | 0.147 |
|  | 192 | **0.152** | 0.180 | 0.167 |
|  | 336 | **0.166** | 0.200 | 0.186 |
|  | 720 | **0.205** | 0.266 | 0.243 |
| Traffic | 96 | 0.401 | **0.398** | 0.402 |
|  | 192 | **0.407** | 0.409 | 0.420 |
|  | 336 | **0.420** | 0.449 | 0.448 |
|  | 720 | **0.456** | 0.589 | 0.539 |
| Weather | 96 | **0.145** | 0.167 | 0.158 |
|  | 192 | **0.188** | 0.229 | 0.211 |
|  | 336 | **0.236** | 0.287 | 0.274 |
|  | 720 | **0.308** | 0.368 | 0.351 |
| ETTm2 | 96 | **0.164** | 0.184 | 0.176 |
|  | 192 | **0.217** | 0.273 | 0.245 |
|  | 336 | **0.269** | 0.309 | 0.295 |
|  | 720 | **0.347** | 0.411 | 0.401 |

Table 11: Results on M4 dataset in SMAPE.

|  | FITS | DLinear | TimesNet | N-Hits | N-Beats |
|---------|------|---------|----------|--------|---------|
| Yearly | 14.00 | 16.96 | 13.38 | 13.41 | 13.43 |
| Quarterly | 10.72 | 12.14 | 10.1 | 10.2 | 10.12 |
| Monthly | 13.49 | 13.51 | 12.67 | 12.7 | 12.67 |

12, 16, and 36 for yearly, quarterly, and monthly prediction accordingly, which is twice the length of the forecasting horizon. Such a short look-back window is very difficult to extract meaningful frequency representation, which further worsens the FITS' performance. We compare FITS with lightweight model DLinear (Zeng et al., 2023), state-of-the-art model TimesNet (Wu et al., 2023) and two hierarchical time series modeling model N-Hits (Challu et al., 2023) and N-Beats (Oreshkin et al., 2019).

# E    CASE STUDY ON OTHER DATASETS

We show the parameter table and performance on other datasets below.

## E.1    ETTH1, ETTM1 & M2

Tab.12 shows the corresponding results on ETTh1 dataset with different settings. ETTh1 shows a abnormal behavior since FITS does not benefits form the longer look-back window, i.e. 720. Instead, it achieves the sota performance at look-back window of 360. We also find this phenomenon in the ETTm1 dataset. We attribute this phenomenon to the distribution shift that exist in the datasets. The longer look-back window will introduce more information from a shifted distribution and sabotage the forecasting result.

Tab. 13 shows the parameter count of parameters of FITS with different settings on the ETTm1 & 2 datasets. Tab. 14 and Tab.15 show the corresponding results on ETTm1 and ETTm2 datasets with different settings. Note that FITS constantly achieves SOTA performance on the ETTm2 dataset with under 10k parameters.

Table 12: The results on the ETTh1 dataset. Values are visualized with a green background, where darker background indicates worse performance. The top-5 best results are highlighted with a red background, and the absolute best result is highlighted with **red bold** font. **F** represents supervision on the forecasting task, while **B+F** represents supervision on backcasting and forecasting tasks.

| Horizon | COF/nth Harmonic | 90 F | 90 B+F | 180 F | 180 B+F | 360 F | 360 B+F | 720 F | 720 B+F |
|---|---|---|---|---|---|---|---|---|---|
| | 2 | 0.391210 | 0.391987 | 0.388999 | 0.389400 | 0.396473 | 0.396321 | 0.402964 | 0.403307 |
| | 3 | 0.390959 | 0.389857 | 0.385510 | 0.386088 | 0.380594 | 0.380825 | 0.386660 | 0.387143 |
| 96 | 4 | 0.387971 | 0.388607 | 0.383368 | 0.384021 | 0.377323 | 0.377619 | 0.383072 | 0.383700 |
| | 5 | 0.386867 | 0.387619 | 0.382036 | 0.382446 | 0.374568 | 0.375034 | 0.380786 | 0.381331 |
| | 6 | 0.386679 | 0.386013 | 0.380810 | 0.381219 | **0.372101** | 0.372712 | 0.379216 | 0.379815 |
| | 2 | 0.441343 | 0.442006 | 0.432260 | 0.432547 | 0.426438 | 0.426715 | 0.435860 | 0.440504 |
| | 3 | 0.440369 | 0.440186 | 0.429132 | 0.429812 | 0.411548 | 0.411818 | 0.420554 | 0.422348 |
| 192 | 4 | 0.438746 | 0.439231 | 0.427605 | 0.427898 | 0.409010 | 0.409368 | 0.417933 | 0.418202 |
| | 5 | 0.437784 | 0.438436 | 0.426070 | 0.426605 | 0.406772 | 0.407058 | 0.415545 | 0.416382 |
| | 6 | 0.436423 | 0.437092 | 0.425226 | 0.425557 | **0.404390** | 0.404649 | 0.414150 | 0.415508 |
| | 2 | 0.482840 | 0.483552 | 0.458024 | 0.460152 | 0.448917 | 0.448952 | 0.455474 | 0.462104 |
| | 3 | 0.482419 | 0.482874 | 0.455709 | 0.458133 | 0.433826 | 0.434296 | 0.440474 | 0.447663 |
| 336 | 4 | 0.481796 | 0.482031 | 0.454570 | 0.456531 | 0.431753 | 0.432849 | 0.437971 | 0.447336 |
| | 5 | 0.480466 | 0.480184 | 0.453879 | 0.455248 | 0.430369 | 0.430036 | 0.434967 | 0.440387 |
| | 6 | 0.477769 | 0.478893 | 0.452324 | 0.454378 | **0.427425** | 0.427670 | 0.434515 | 0.439989 |
| | 2 | 0.471969 | 0.474663 | 0.442253 | 0.444961 | 0.445856 | 0.447131 | 0.450286 | 0.451247 |
| | 3 | 0.470309 | 0.472660 | 0.440675 | 0.443468 | 0.430920 | 0.432436 | 0.435924 | 0.436795 |
| 720 | 4 | 0.469793 | 0.471439 | 0.438692 | 0.442282 | 0.428483 | 0.430407 | 0.433446 | 0.438730 |
| | 5 | 0.470391 | 0.471038 | 0.437598 | 0.441572 | 0.426720 | 0.428304 | 0.431934 | 0.433103 |
| | 6 | 0.468232 | 0.469044 | 0.438001 | 0.440176 | **0.424749** | 0.425963 | 0.430258 | 0.431978 |

Table 13: The number of parameters under different settings on ETTm1 & ETTm2 dataset.

| Horizon | COF/nth Harmonic | 90 | 180 | 360 | 720 |
|---|---|---|---|---|---|
| | 4 | 420 | 513 | 621 | 1330 |
| | 6 | 561 | 759 | 1015 | 2444 |
| 96 | 8 | 703 | 1053 | 1505 | 3835 |
| | 10 | 861 | 1426 | 2050 | 5609 |
| | 12 | 1035 | 1820 | 2726 | 7636 |
| | 14 | 1225 | 2262 | 5561 | 16974 |
| | 4 | 645 | 703 | 759 | 1505 |
| | 6 | 850 | 1035 | 1218 | 2726 |
| 192 | 8 | 1064 | 1431 | 1820 | 4307 |
| | 10 | 1302 | 1922 | 2501 | 6248 |
| | 12 | 1564 | 2450 | 3290 | 8549 |
| | 14 | 1875 | 3042 | 6767 | 18942 |
| | 4 | 990 | 969 | 966 | 1715 |
| | 6 | 1275 | 1449 | 1566 | 3149 |
| 336 | 8 | 1615 | 1998 | 2275 | 5015 |
| | 10 | 1974 | 2666 | 3157 | 7242 |
| | 12 | 2392 | 3395 | 4136 | 9960 |
| | 14 | 2825 | 4212 | 8509 | 21894 |
| | 4 | 1890 | 1710 | 1518 | 2380 |
| | 6 | 2448 | 2530 | 2436 | 4324 |
| 720 | 8 | 3078 | 3510 | 3570 | 6844 |
| | 10 | 3780 | 4650 | 4920 | 9940 |
| | 12 | 4554 | 5950 | 6486 | 13612 |
| | 14 | 5400 | 7410 | 13266 | 30012 |

## E.2 TRAFFIC

Tab. 16 shows the parameter count of parameters of FITS with different settings on the Traffic dataset. Tab. 17shows the result on the Traffic dataset with different settings correspondingly. The traffic dataset has a very large amount of channels, making many models need many parameters to model the temporal information. FITS only needs 50k parameters to achieve comparable performance.

Table 14: The results on the ETTm1 dataset. Values are visualized with a green background, where darker background indicates worse performance. The top-5 best results are highlighted with a red background, and the absolute best result is highlighted with **red bold** font. **F** represents supervision on the forecasting task, while **B+F** represents supervision on backcasting and forecasting tasks.

| Horizon | Look-back Window | 90 | | 180 | | 360 | | 720 | |
|---|---|---|---|---|---|---|---|---|---|
| | COF/nth Harmonic | **F** | **B+F** | **F** | **B+F** | **F** | **B+F** | **F** | **B+F** |
| 96 | 6 | 0.365445 | 0.364920 | 0.312641 | 0.312953 | 0.305685 | 0.306146 | 0.314674 | 0.314514 |
| | 8 | 0.364851 | 0.364060 | 0.312192 | 0.312162 | 0.304728 | 0.304859 | 0.311096 | 0.311866 |
| | 10 | 0.364031 | 0.363901 | 0.311927 | 0.312576 | 0.303561 | 0.304319 | 0.310334 | 0.310245 |
| | 12 | 0.363204 | 0.363368 | 0.311663 | 0.311027 | 0.303412 | 0.303837 | 0.309762 | 0.309719 |
| | 14 | 0.362444 | 0.362795 | 0.311292 | 0.311589 | **0.303153** | 0.303350 | 0.309248 | 0.309280 |
| 192 | 6 | 0.400734 | 0.400507 | 0.348337 | 0.348248 | 0.339300 | 0.339427 | 0.342016 | 0.342371 |
| | 8 | 0.400492 | 0.400347 | 0.347860 | 0.347737 | 0.338135 | 0.338368 | 0.339153 | 0.339648 |
| | 10 | 0.399691 | 0.399442 | 0.347713 | 0.348003 | 0.337680 | 0.337875 | 0.338802 | 0.338988 |
| | 12 | 0.399838 | 0.398958 | 0.347586 | 0.347586 | 0.337414 | 0.337599 | 0.338334 | 0.338354 |
| | 14 | 0.399357 | 0.399249 | 0.347410 | 0.347304 | **0.337172** | 0.337290 | 0.337920 | 0.338687 |
| 336 | 6 | 0.431620 | 0.431774 | 0.384310 | 0.384216 | 0.372982 | 0.373155 | 0.369938 | 0.369875 |
| | 8 | 0.431072 | 0.431733 | 0.384172 | 0.384051 | 0.371957 | 0.372328 | 0.367226 | 0.367214 |
| | 10 | 0.431126 | 0.431569 | 0.383794 | 0.383776 | 0.371696 | 0.371917 | 0.366873 | 0.367586 |
| | 12 | 0.430388 | 0.431283 | 0.383467 | 0.383656 | 0.371289 | 0.371662 | 0.366383 | 0.366726 |
| | 14 | 0.430749 | 0.431064 | 0.383719 | 0.383675 | 0.371363 | 0.371460 | **0.366238** | 0.366492 |
| 720 | 6 | 0.492552 | 0.492473 | 0.443358 | 0.443911 | 0.427787 | 0.428065 | 0.418358 | 0.418436 |
| | 8 | 0.491822 | 0.492750 | 0.443295 | 0.443706 | 0.427054 | 0.427269 | 0.415964 | 0.416064 |
| | 10 | 0.492088 | 0.492691 | 0.443319 | 0.443260 | 0.426868 | 0.427377 | 0.415702 | 0.416024 |
| | 12 | 0.491904 | 0.492589 | 0.443103 | 0.443476 | 0.426748 | 0.426819 | 0.415402 | 0.415881 |
| | 14 | 0.491202 | 0.491228 | 0.443076 | 0.443384 | 0.426571 | 0.427071 | **0.415266** | 0.415599 |

Table 15: The results on the ETTm2 dataset. Values are visualized with a green background, where darker background indicates worse performance. The top-5 best results are highlighted with a red background, and the absolute best result is highlighted with **red bold** font. **F** represents supervision on the forecasting task, while **B+F** represents supervision on backcasting and forecasting tasks.

| Horizon | Look-back Window | 90 | | 180 | | 360 | | 720 | |
|---|---|---|---|---|---|---|---|---|---|
| | COF/nth Harmonic | **F** | **B+F** | **F** | **B+F** | **F** | **B+F** | **F** | **B+F** |
| 96 | 6 | 0.185981 | 0.185774 | 0.174363 | 0.174277 | 0.166817 | 0.166589 | 0.164719 | 0.164500 |
| | 8 | 0.185601 | 0.185615 | 0.174002 | 0.173797 | 0.166025 | 0.165700 | 0.164127 | 0.163744 |
| | 10 | 0.185515 | 0.185338 | 0.173717 | 0.173536 | 0.165634 | 0.165413 | 0.163178 | 0.162928 |
| | 12 | 0.185489 | 0.185038 | 0.173951 | 0.173368 | 0.165369 | 0.165113 | 0.162719 | 0.162550 |
| | 14 | 0.185646 | 0.185048 | 0.173502 | 0.173315 | 0.165362 | 0.165198 | 0.162575 | **0.162346** |
| 192 | 6 | 0.249365 | 0.249119 | 0.233717 | 0.233428 | 0.221364 | 0.220948 | 0.218563 | 0.218267 |
| | 8 | 0.249366 | 0.248945 | 0.233405 | 0.233171 | 0.220596 | 0.220165 | 0.218115 | 0.217808 |
| | 10 | 0.248919 | 0.248686 | 0.233276 | 0.232929 | 0.220104 | 0.220518 | 0.217338 | 0.216963 |
| | 12 | 0.248677 | 0.248890 | 0.233005 | 0.233042 | 0.220007 | 0.219895 | 0.216927 | 0.216714 |
| | 14 | 0.248678 | 0.248454 | 0.233162 | 0.232763 | 0.219897 | 0.219597 | 0.216879 | **0.216650** |
| 336 | 6 | 0.309083 | 0.308863 | 0.286387 | 0.286170 | 0.273920 | 0.273816 | 0.269833 | 0.269620 |
| | 8 | 0.309234 | 0.308577 | 0.286186 | 0.286041 | 0.273418 | 0.273171 | 0.269393 | 0.269252 |
| | 10 | 0.308768 | 0.308713 | 0.286102 | 0.285768 | 0.273038 | 0.272893 | 0.268921 | 0.268596 |
| | 12 | 0.308741 | 0.308568 | 0.286529 | 0.285881 | 0.272931 | 0.272763 | 0.268468 | 0.268273 |
| | 14 | 0.308759 | 0.308393 | 0.286232 | 0.285711 | 0.272905 | 0.272804 | 0.268366 | **0.268219** |
| 720 | 6 | 0.408977 | 0.408844 | 0.384164 | 0.383995 | 0.366645 | 0.366654 | 0.350173 | 0.349770 |
| | 8 | 0.409260 | 0.408714 | 0.383980 | 0.383899 | 0.366070 | 0.366085 | 0.349659 | 0.349619 |
| | 10 | 0.408793 | 0.408703 | 0.383886 | 0.383827 | 0.365935 | 0.365935 | 0.349019 | 0.348881 |
| | 12 | 0.408698 | 0.408639 | 0.383921 | 0.383664 | 0.365810 | 0.365805 | 0.348831 | 0.348863 |
| | 14 | 0.408765 | 0.408479 | 0.383926 | 0.383620 | 0.365790 | 0.365801 | 0.348938 | **0.348766** |

## E.3 WEATHER

Tab. 18 shows the parameter count of parameters of FITS with different settings on the Weather dataset. Tab. 17shows the result on the Traffic dataset with different settings correspondingly. Note that we achieve the result in the main table by setting the COF as 75 and the look-back window as 700.

Table 16: The number of parameters under different settings on Traffic dataset.

| Horizon | COF/nth Harmonic | 90 | 180 | 360 | 720 |
|---------|------------------|-----|------|------|------|
| | | \multicolumn{4}{c}{Look-back Window} |
| 96 | 3 | 1035 | 1820 | 4307 | 12064 |
| | 5 | 1922 | 3876 | 10374 | 31042 |
| | 8 | 3698 | 8475 | 24186 | 75628 |
| | 10 | N/A | 12558 | 36765 | 116202 |
| 192 | 3 | 1564 | 2450 | 5192 | 13520 |
| | 5 | 2914 | 5253 | 12558 | 34694 |
| | 8 | 5633 | 11400 | 29329 | 84434 |
| | 10 | N/A | 16926 | 44460 | 130005 |
| 336 | 3 | 2392 | 3395 | 6608 | 15704 |
| | 5 | 4402 | 7293 | 15834 | 40006 |
| | 8 | 8514 | 15900 | 36974 | 97902 |
| | 10 | N/A | 23478 | 56088 | 150549 |
| 720 | 3 | 4554 | 5950 | 10266 | 21424 |
| | 5 | 8370 | 12750 | 24570 | 54780 |
| | 8 | 16254 | 27750 | 57546 | 133644 |
| | 10 | N/A | 40950 | 87210 | 205440 |

Table 17: The results on the Traffic dataset. Values are visualized with a green background, where darker background indicates worse performance. The top-5 best results are highlighted with a red background, and the absolute best result is highlighted with **red bold** font. **F** represents supervision on the forecasting task, while **B+F** represents supervision on backcasting and forecasting tasks.

| Horizon | COF/nth Harmonic | 90 F | 90 B+F | 180 F | 180 B+F | 360 F | 360 B+F | 720 F | 720 B+F |
|---------|------------------|------|--------|-------|---------|-------|---------|-------|---------|
| | | \multicolumn{8}{c}{Look-back Window} |
| 96 | 3 | 0.694065 | 0.694425 | 0.474606 | 0.475881 | 0.455815 | 0.457292 | 0.436317 | 0.436616 |
| | 5 | 0.686110 | 0.684290 | 0.457057 | 0.456547 | 0.419903 | 0.419748 | 0.397558 | 0.397210 |
| | 8 | 0.682876 | 0.681464 | 0.452024 | 0.451470 | 0.410857 | 0.410458 | 0.387791 | 0.388830 |
| | 10 | N/A | N/A | 0.451340 | 0.450850 | 0.409948 | 0.409614 | **0.385763** | 0.386596 |
| 192 | 3 | 0.627212 | 0.636434 | 0.481686 | 0.485085 | 0.463516 | 0.464170 | 0.442661 | 0.443547 |
| | 5 | 0.622882 | 0.622451 | 0.467921 | 0.467470 | 0.431367 | 0.431097 | 0.407908 | 0.407850 |
| | 8 | 0.620314 | 0.620003 | 0.463119 | 0.462929 | 0.422517 | 0.422336 | 0.399032 | 0.399101 |
| | 10 | N/A | N/A | 0.462595 | 0.461907 | 0.421705 | 0.421677 | **0.397286** | 0.398034 |
| 336 | 3 | 0.635301 | 0.662283 | 0.496200 | 0.510793 | 0.473090 | 0.476491 | 0.454243 | 0.456989 |
| | 5 | 0.632244 | 0.631760 | 0.481267 | 0.480599 | 0.442476 | 0.442128 | 0.420268 | 0.420239 |
| | 8 | 0.629962 | 0.629700 | 0.477111 | 0.476673 | 0.434504 | 0.434124 | 0.411608 | 0.412114 |
| | 10 | N/A | N/A | 0.476248 | 0.476044 | 0.433656 | 0.433456 | 0.410500 | **0.410417** |
| 720 | 3 | 0.685472 | 0.732168 | 0.529004 | 0.606921 | 0.500635 | 0.587891 | 0.488116 | 0.489934 |
| | 5 | 0.670385 | 0.669979 | 0.507742 | 0.507104 | 0.469442 | 0.469337 | 0.456470 | 0.456207 |
| | 8 | 0.668054 | 0.668322 | 0.504645 | 0.536565 | 0.463208 | 0.463744 | 0.449778 | 0.449220 |
| | 10 | N/A | N/A | 0.503795 | 0.503702 | 0.462643 | 0.463091 | 0.448882 | **0.448182** |

## E.4 ELECTRICITY

Tab. 20 shows the parameter count of parameters of FITS with different settings on the Electricity dataset. Tab. 21 shows the result on the Electricity dataset with different settings correspondingly. We find that the Electricity dataset is sensitive to the COF. This is because this dataset shows significant multi-periodicity, which requires capturing high-frequency components. Otherwise, FITS will not learn such information.

Table 18: The number of parameters per channel under different settings on Weather dataset.

| Horizon | COF/nth Harmonic | Look-back Window | | | |
|---|---|---|---|---|---|
| | | 90 | 180 | 360 | 720 |
| 96 | 5 | 496 | 630 | 806 | 1845 |
| | 8 | 703 | 1053 | 1505 | 3835 |
| | 10 | 861 | 1426 | 2050 | 5609 |
| | 12 | 1035 | 1820 | 2726 | 7636 |
| 192 | 5 | 752 | 861 | 988 | 2050 |
| | 8 | 1064 | 1431 | 1820 | 4307 |
| | 10 | 1302 | 1922 | 2501 | 6248 |
| | 12 | 1564 | 2450 | 3290 | 8549 |
| 336 | 5 | 1136 | 1197 | 1248 | 2378 |
| | 8 | 1615 | 1998 | 2275 | 5015 |
| | 10 | 1974 | 2666 | 3157 | 7242 |
| | 12 | 2392 | 3395 | 4136 | 9960 |
| 720 | 5 | 2160 | 2100 | 1950 | 3280 |
| | 8 | 3078 | 3510 | 3570 | 6844 |
| | 10 | 3780 | 4650 | 4920 | 9940 |
| | 12 | 4554 | 5950 | 6486 | 13612 |

Table 19: The results on the Weather dataset. Values are visualized with a green background, where darker background indicates worse performance. The top-5 best results are highlighted with a red background, and the absolute best result is highlighted with **red bold** font. **F** represents supervision on the forecasting task, while **B+F** represents supervision on backcasting and forecasting tasks.

| Horizon | COF/nth Harmonic | Look-back Window | | | | | | | |
|---|---|---|---|---|---|---|---|---|---|
| | | 90 | | 180 | | 360 | | 720 | |
| | | F | B+F | F | B+F | F | B+F | F | B+F |
| 96 | 5 | 0.168999 | 0.167446 | 0.154289 | 0.153895 | 0.146525 | 0.147573 | 0.145413 | 0.145591 |
| | 8 | 0.168489 | 0.167466 | 0.154478 | 0.152689 | 0.150962 | 0.145706 | 0.150321 | 0.144873 |
| | 10 | 0.167841 | 0.16671 | 0.157372 | 0.154233 | 0.144939 | 0.145926 | 0.14378 | 0.143663 |
| | 12 | 0.167211 | 0.167596 | 0.153383 | 0.152575 | 0.145388 | 0.144406 | 0.144109 | **0.143549** |
| 192 | 5 | 0.216142 | 0.214978 | 0.199501 | 0.20005 | 0.190332 | 0.188983 | 0.188193 | 0.187653 |
| | 8 | 0.215683 | 0.214616 | 0.2003 | 0.197924 | 0.190594 | 0.188961 | 0.187488 | 0.187293 |
| | 10 | 0.216368 | 0.215074 | 0.198559 | 0.196907 | 0.188374 | 0.188846 | 0.189149 | 0.187211 |
| | 12 | 0.215767 | 0.214471 | 0.19829 | 0.196227 | 0.188737 | 0.188946 | **0.186627** | 0.187315 |
| 336 | 5 | 0.271176 | 0.268613 | 0.252509 | 0.250649 | 0.2425 | 0.241132 | 0.238817 | 0.237748 |
| | 8 | 0.271297 | 0.269261 | 0.254065 | 0.251179 | 0.242187 | 0.241113 | 0.237223 | **0.236302** |
| | 10 | 0.270489 | 0.26817 | 0.25271 | 0.25147 | 0.241169 | 0.241497 | 0.237585 | 0.23675 |
| | 12 | 0.270616 | 0.268199 | 0.252692 | 0.250244 | 0.241046 | 0.240129 | 0.236639 | 0.236732 |
| 720 | 5 | 0.350177 | 0.347762 | 0.331098 | 0.329988 | 0.317609 | 0.317085 | 0.307792 | 0.308279 |
| | 8 | 0.350805 | 0.348277 | 0.332104 | 0.329343 | 0.317169 | 0.316756 | 0.307681 | 0.307556 |
| | 10 | 0.350146 | 0.347992 | 0.331052 | 0.329464 | 0.317204 | 0.316535 | 0.307876 | 0.307997 |
| | 12 | 0.349919 | 0.347507 | 0.330589 | 0.328929 | 0.31728 | 0.317051 | **0.307549** | 0.307695 |

Table 20: The number of parameters under different settings on Electricity dataset.

| Horizon | COF/nth Harmonic | Look-back Window | | | |
|---------|------------------|------|------|------|------|
| | | **90** | **180** | **360** | **720** |
| 96 | **4** | 1431 | 2752 | 6975 | 20385 |
| | **6** | 2450 | 5192 | 14338 | 43734 |
| | **8** | 3698 | 8475 | 24186 | 75628 |
| | **10** | N/A | 12558 | 36765 | 116202 |
| 192 | **4** | 2187 | 3698 | 8475 | 22815 |
| | **6** | 3710 | 7021 | 17334 | 48856 |
| | **8** | 5633 | 11400 | 29329 | 84434 |
| | **10** | N/A | 16926 | 44460 | 130005 |
| 336 | **4** | 3321 | 5160 | 10725 | 26460 |
| | **6** | 5600 | 9794 | 21828 | 56539 |
| | **8** | 8514 | 15900 | 36974 | 97902 |
| | **10** | N/A | 23478 | 56088 | 150549 |
| 720 | **4** | 6318 | 9030 | 16650 | 36180 |
| | **6** | 10710 | 17110 | 34026 | 77224 |
| | **8** | 16254 | 27750 | 57546 | 133644 |
| | **10** | N/A | 40950 | 87210 | 205440 |

Table 21: The results on the Electricity dataset. Values are visualized with a green background, where darker background indicates worse performance. The top-5 best results are highlighted with a red background, and the absolute best result is highlighted with **red bold** font. **F** represents supervision on the forecasting task, while **B+F** represents supervision on backcasting and forecasting tasks.

| Horizon | COF/nth Harmonic | Look-back Window | | | | | | | |
|---------|------------------|---------|---------|---------|---------|---------|---------|---------|---------|
| | | 90 | | 180 | | 360 | | 720 | |
| | | **F** | **B+F** | **F** | **B+F** | **F** | **B+F** | **F** | **B+F** |
| 96 | 4 | 0.211863 | 0.211541 | 0.164613 | 0.164616 | 0.156359 | 0.156297 | 0.150566 | 0.150438 |
| | 6 | 0.207940 | 0.207753 | 0.158720 | 0.158540 | 0.145222 | 0.145141 | 0.141353 | 0.141240 |
| | 8 | 0.205725 | 0.205731 | 0.155645 | 0.155360 | 0.141808 | 0.141797 | 0.136764 | 0.136518 |
| | 10 | N/A | N/A | 0.153865 | 0.153727 | 0.140404 | 0.140146 | 0.134548 | **0.134512** |
| 192 | 4 | 0.208883 | 0.208732 | 0.177320 | 0.177169 | 0.170017 | 0.169900 | 0.164768 | 0.164958 |
| | 6 | 0.204794 | 0.204745 | 0.171682 | 0.171478 | 0.159142 | 0.158998 | 0.155794 | 0.155677 |
| | 8 | 0.202812 | 0.202812 | 0.168498 | 0.168369 | 0.155829 | 0.155772 | 0.151284 | 0.151151 |
| | 10 | N/A | N/A | 0.166973 | 0.166850 | 0.154208 | 0.154133 | 0.149191 | **0.149113** |
| 336 | 4 | 0.223752 | 0.223654 | 0.193835 | 0.193764 | 0.185941 | 0.185745 | 0.180178 | 0.180076 |
| | 6 | 0.225369 | 0.228277 | 0.188469 | 0.188316 | 0.175312 | 0.175207 | 0.171449 | 0.171408 |
| | 8 | 0.217910 | 0.217896 | 0.185530 | 0.185429 | 0.172143 | 0.172221 | 0.167087 | 0.167156 |
| | 10 | N/A | N/A | 0.184023 | 0.183940 | 0.170639 | 0.170568 | **0.165106** | 0.165353 |
| 720 | 4 | 0.264973 | 0.264844 | 0.232324 | 0.231985 | 0.223079 | 0.222988 | 0.217765 | 0.217644 |
| | 6 | 0.262983 | 0.262587 | 0.227060 | 0.227014 | 0.213388 | 0.213266 | 0.209735 | 0.209591 |
| | 8 | 0.261890 | 0.262908 | 0.224484 | 0.224334 | 0.210542 | 0.210451 | 0.205780 | 0.205604 |
| | 10 | N/A | N/A | 0.223122 | 0.223114 | 0.209192 | 0.209104 | 0.204054 | **0.203816** |

# F  FULL ANOMALY DETECTION RESULTS

The full results with Accuracy, Precision, Recall, and F1-score are shown in Tab. 22. For better performance, we also conduct experiments only on the first channel of the SML dataset, denoted as (C0). We also trained FITS using only the analog channels of SWaT, denoted as (analog).

Table 22: Full results on five datasets.

| Datasets | Accuracy | Precision | Recall | F1-score |
|----------|----------|-----------|--------|----------|
| SMD | 99.92 | 99.9 | 100 | 99.95 |
| PSM | 94.43 | 97.2 | 90.43 | 93.69 |
| SWaT | 99.42 | 97.84 | 100 | 98.9 |
| SWaT(analog) | 97.81 | 91.74 | 100 | 95.69 |
| SMAP | 89.39 | 77.52 | 65.05 | 70.74 |
| MSL | 81.52 | 61.38 | 80.16 | 69.52 |
| MSL(C0) | 83.77 | 81.34 | 75.15 | 78.12 |

# G  ANOMALY DETECTION RESULTS ON SYNTHETIC DATASET

We generate the synthetic dataset using the script provided in the benchmark with the default setting, i.e., 5% outlier on each channel with different outlier types. We generate 4000 time-steps as our

| Model | Precision | Recall | F1-score |
|---|---|---|---|
| FITS-win24 | 1 | 1 | 1 |
| FITS-win50 | 1 | 1 | 1 |
| FITS-win100 | 1 | 0.9993 | 0.9996 |
| FITS-win400 | 1 | 0.9991 | 0.9995 |
| AR | 0.59 | 0.77 | 0.64 |
| GBRT | 0.47 | 0.56 | 0.51 |
| LSTM-RNN | 0.22 | 0.26 | 0.24 |
| IForest | 0.48 | 0.57 | 0.52 |
| OCSVM | 0.62 | 0.74 | 0.67 |
| AutoEncoder | 0.20 | 0.24 | 0.22 |
| GAN | 0.15 | 0.15 | 0.15 |

Table 23: Results on the synthetic dataset.

dataset, in which we take 2500 for training and the rest 1500 for testing. For our FITS model, we use four different reconstruction windows, labeled as FITS-winxxx. We compare with the results retrieved from Table 17 of the original paper (Lai et al., 2021). The result is shown in 23.

## H    DATASETS VISUALIZATION ON ANOMALY DETECTION

As shown in Fig. 6 and Fig. 7, most PSM and SMD datasets channels are analog values. Especially the PSM dataset shows great periodicity.

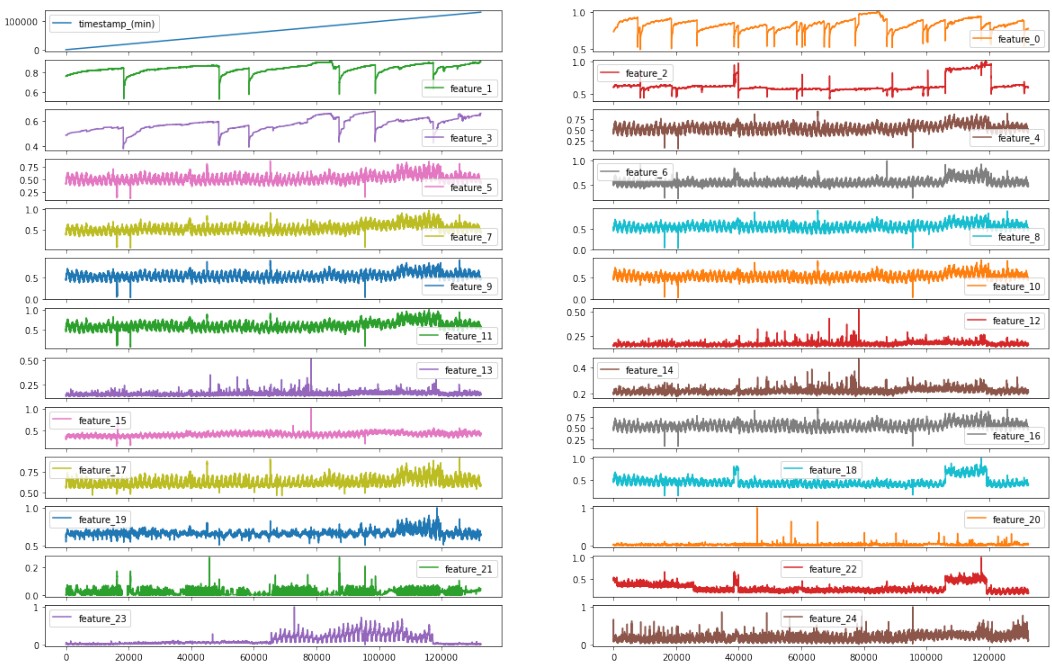

Figure 6: Waveform of PSM dataset.

While some channels in the SWaT dataset are binary event values, as shown in Fig. 8.

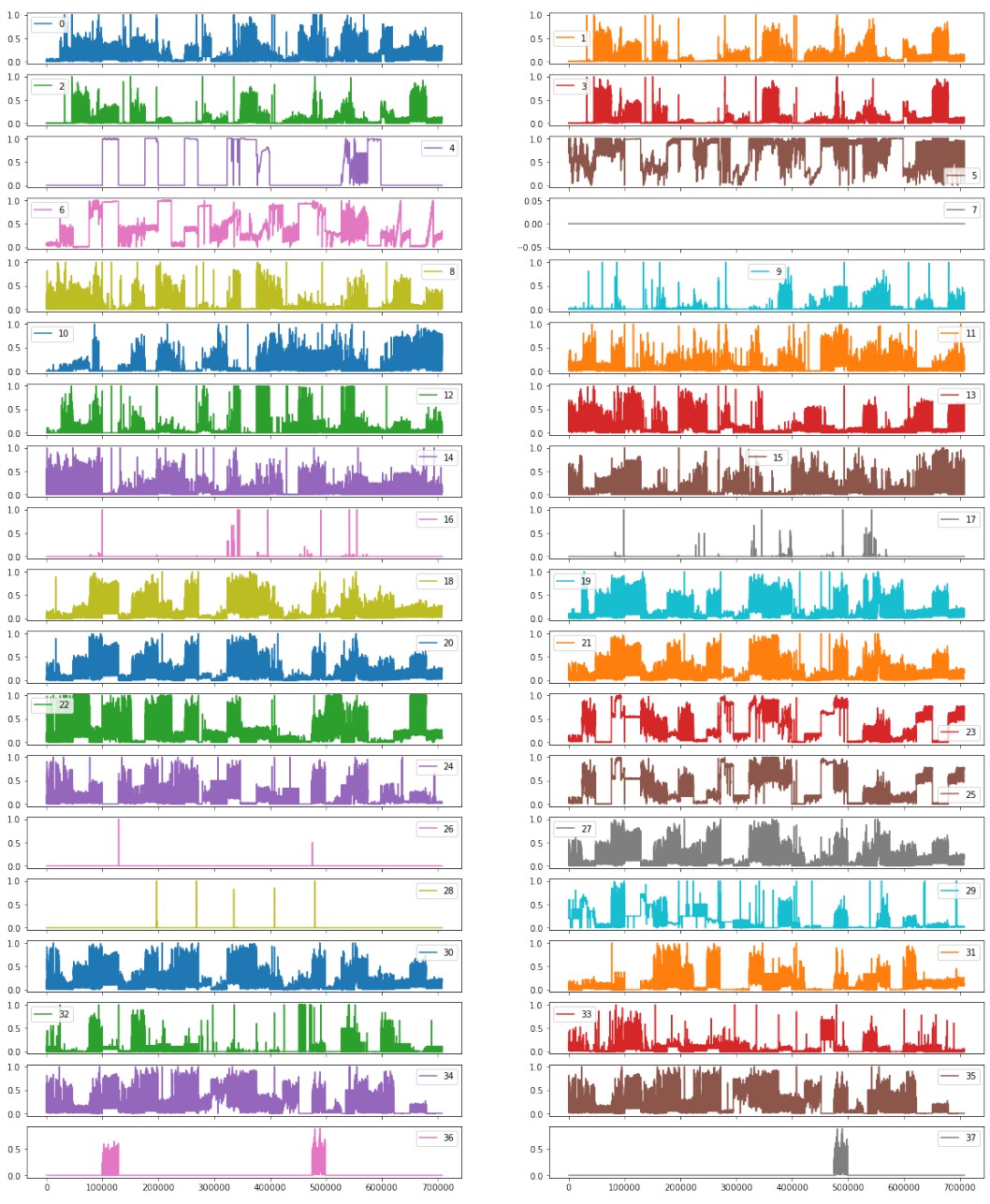

Figure 7: Waveform of SMD dataset.

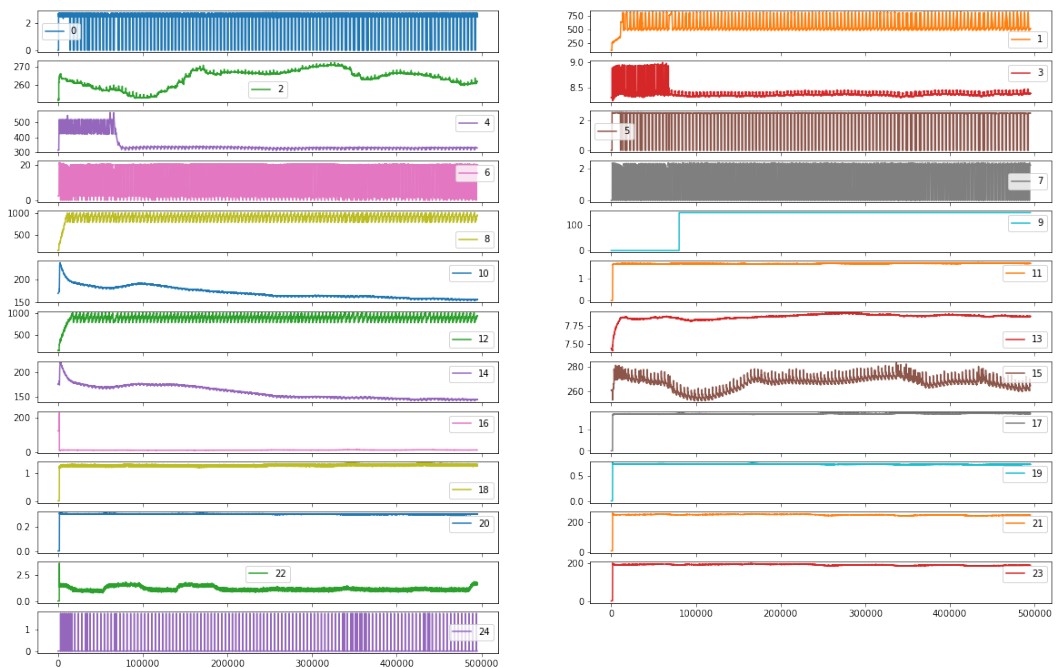

Figure 8: Waveform of SWAT dataset.

However, as shown in Fig. 9 and Fig. 10, for SMAP and MSL datasets, most channels are binary event values that are hard for FITS to learn frequency representation.

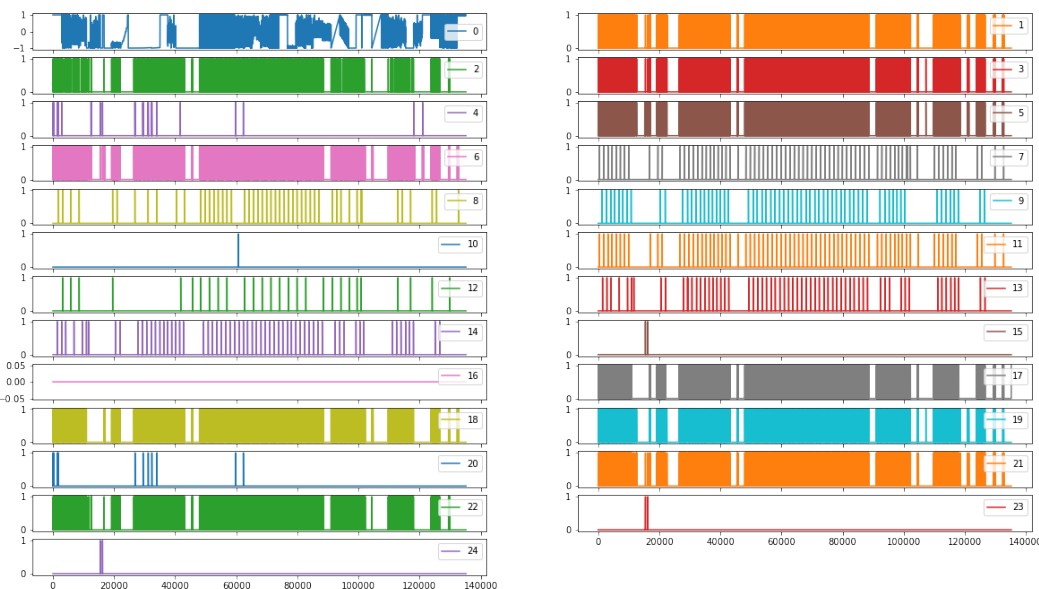

Figure 9: Waveform of SMAP dataset.

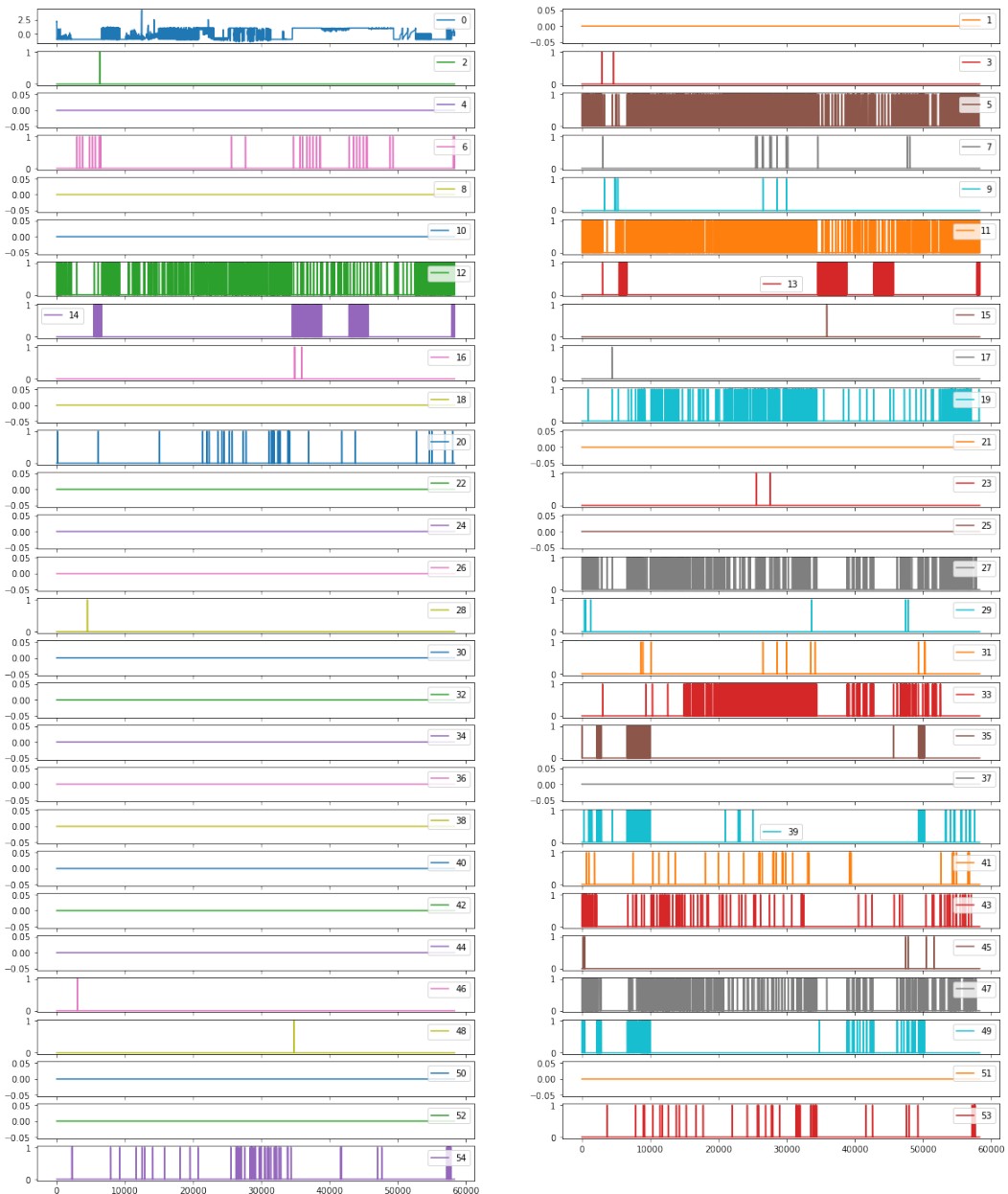

Figure 10: Waveform of MSL dataset.

# I PARAMETER COUNTS FOR ANOMALY DETECTION

We use a fixed sliding window of 200 and 400 for all the datasets and do not apply any frequency filter. The downsample rate is set as 4 for any dataset. Thus, the number of parameters is as Tab. 24.

Table 24: MACs and parameter count of FITS on Anomaly Detection task. We report the MACs on the SWaT dataset which has 55 channels.

| Window | Params | MACs |
|--------|--------|--------|
| 200 | 2600 | 137.5k |
| 400 | 10200 | 550k |

