# OpenReview forum: "FITS: Modeling Time Series with $10k$ Parameters"
_ICLR.cc/2024/Conference — ICLR 2024 spotlight_

### Official Review · Reviewer_u3QE · 2023-10-30

**Soundness:** 3 good
**Presentation:** 3 good
**Contribution:** 3 good
**Rating:** 8
**Confidence:** 2

**Summary:**

Modern time series forecasting methods are heavily-parametrised, and in spite of their compelling performance on benchmark datasets, may not be appropriate in resource-constrained settings. In the spirit of recent work leveraging representations of time series in the frequency domain, the authors propose a new methodology that captures amplitude and phase information using a real-valued neural network. A low-pass filter is also included to further reduce the size of the model. An extensive experimental evaluation demonstrates how model performance is comparable (and sometimes better) to other SOTA techniques while also having orders of magnitude less parameters. The suitability of the model to anomaly detection via time series reconstruction is also validated on a variety of datasets.

**Strengths:**

- The reduction in model size enabled by the proposed architecture is commendable, and the minor trade-off with predictive performance (if any at all in some cases) makes this a compelling model for practitioners working with time series forecasting models.
- The paper is well-written and nicely structured. Although there has been a recent resurgence in interest on how analysis in the frequency domain can be applied to forecasting, the authors adequately position this paper in the context of related work.
- The experimental evaluation is extensive, and I especially appreciated the extension to anomaly detection, which is one of the more common downstream tasks stemming from time series forecasting. The insights on why the model works better on some datasets than others (e.g. citing the binary nature of some time series, as well as overall limitations with benchmarks in the field) was also appreciated.

**Weaknesses:**

- There are quite a few bits and pieces to the model pipeline, and at times the extent to which each component contributes towards overall performance is unclear. Perhaps a few ablation studies in the supplementary material can address this?
- The concluding remarks of the paper focus on a few settings where the method may not be most appropriate, but I would have liked to see broader thoughts on future work, such as further reductions to model size, and possibly interpretability.
- While I appreciated the experiment on anomaly detection, this is a field of study where (as referenced by the authors themselves), a few metrics may not be sufficient for fully capturing the success of different techniques. There is consequently a risk that the experiment featured here might not be sufficiently conclusive (although the results obtained on synthetic data are a step in the right direction).

**Questions:**

Please refer to comments in the section on *Weaknesses* as a guide for the rebuttal.

---

> ### Author Response · Authors · 2023-11-18
>
> Thanks for your insightful questions.
>
> - There are quite a few bits and pieces to the model pipeline, and at times the extent to which each component contributes towards overall performance is unclear. Perhaps a few ablation studies in the supplementary material can address this?
>
>     A: *We understand that ablation study is very important to understand the impact of each part. However, **FITS is a very compact model. Removing any of its part will make it not functional. And FITS only have one hyperparameter which is the interpolation rate. We have done comprehensive ablation studies on the cutoff frequency, please see the Tab.5 of the manuscript and Section D of the appendix.***
>
> - The concluding remarks of the paper focus on a few settings where the method may not be most appropriate, but I would have liked to see broader thoughts on future work, such as further reductions to model size, and possibly interpretability.
>
>     A: *Thank you for your feedback. You've raised valuable points. In response to your query about future work, we have plans to delve deeper into the exploration of complex-valued neural networks in the frequency domain, including the investigation of complex-valued attention mechanisms. Regarding interpretability, we acknowledge its importance and are actively working on improving our visualization techniques to make the results more interpretable.*
>
>     *I'd like to highlight that we've included an interactive notebook in the anonymous code repository, which can assist in understanding how FITS learns various frequency combinations in the frequency domain. As for further model size reduction, we understand that it can be a challenging task, and we are currently in the process of evaluating its feasibility.*
>
>     *We have add the discussion on the future work to the manuscript according to your advice.*
>
>
> - While I appreciated the experiment on anomaly detection, this is a field of study where (as referenced by the authors themselves), a few metrics may not be sufficient for fully capturing the success of different techniques. There is consequently a risk that the experiment featured here might not be sufficiently conclusive (although the results obtained on synthetic data are a step in the right direction).
>
>     A: *We acknowledge that relying solely on simple metrics may not adequately capture the performance of FITS in real-world tasks. However, to ensure a fair comparison, we have followed the approach of previous works, specifically the Anomaly Transformer, to demonstrate the superior performance of FITS. Nevertheless, we recognize the importance of interpretability and plan to delve deeper into this aspect by investigating the interpretability of FITS using synthetic datasets. For a more comprehensive visualization and interpretation, we invite you to explore the jupyter notebook available in our anonymous repository.*

---

> > ### Comment · Reviewer_u3QE · 2023-11-22
> > **Acknowledgement of Rebuttal**
> >
> > Thank you for your reply - I have also followed the discussion on the other reviews, and decided to increase my score to 8.

---

### Official Review · Reviewer_r1EX · 2023-10-31

**Soundness:** 3 good
**Presentation:** 3 good
**Contribution:** 3 good
**Rating:** 8
**Confidence:** 3

**Summary:**

The authors propose a parameter-efficient model architecture for time series anomaly detection and forecasting. The model's pipeline consists of the Fourier transform of the original time series, followed by a low-pass filter in the freq. domain, a linear layer (complex-valued), padding, and the inverse transform to map back into the time domain. For forecasting, the result of the inverse transform can be used directly, while for anomaly detection, a reconstruction error threshold, which increases the F1-score, is determined on the validation set. The authors show competitive performance on various benchmark data sets.

**Strengths:**

The manuscript is well-organized and clearly written. I appreciate the preliminary section, which is very instructive. The experimental evaluation is thorough, and many competing methods are compared in a rigorous manner. Lastly, I appreciate the different ablation studies the authors performed. Overall, the results are convincing, and I believe the proposed method sets a great benchmark for such a parameter-efficient method.

**Weaknesses:**

The presentation of FITS could be more comprehensive (more details in the questions). Furthermore, I would like to see more "basic" baselines (e.g., ARIMA, DeepAR for forecasting, and Random Cut Forest for Anomaly detection). To further substantiate the claims about performance (and to put them on a statistically sound foundation), a critical difference plot[1] may be a useful analysis tool. Lastly, the method employs several heuristics that need to be tuned in the real world, a drawback it shares with other methods.

1. Demšar, J. (2006). Statistical comparisons of classifiers over multiple data sets. The Journal of Machine learning research, 7, 1-30.


**Rebuttal:** I increased my score by one after the rebuttal, as all my concerns have been addressed appropriately.

**Questions:**

* Can you explain why $x(t-\tau)$ is a shift forward in time, rather than backward?
* Can you add a more thorough caption to Fig. 2? As it stands, it is not comprehensible enough to fully understand your method. Is it just a single linear layer/what is the output dimension? What are the typical dimensions of $\tilde{X}$? Are (i)RIN and (i)rFFT performed in parallel or sequentially? Please elaborate more on how RIN works, either in the Figure or its caption.
* Have you experimented with more sophisticated layers than just linear?
* What is the back-casting task, and how is it used in training?
* Can you motivate in 1-2 sentences why the output length needs to be controlled (Section 3.3)?
* You say the LPF discards components beyond the model's learning capabilities. What do you mean by this? The model's capabilities are independent of the signal, and high frequencies are not always noise or irrelevant. It seems domain-specific. Furthermore, you say high-frequency components such as trends are filtered out. First, trend signals can be low-frequency, and second, they are particularly important in the forecasting realm.
* The method is particularly interesting because of its parameter-effectiveness. Can you perform experiments on real-time inference time on simple hardware (not necessarily edge devices) to substantiate this (e.g., real-time anomaly detection)? What are the challenges regarding domain shifts and refitting, and how could they be addressed? This could substantially improve the focus and contribution of the paper.

---

> ### Author Response · Authors · 2023-11-18
>
> Thanks for your insightful questions. We will address your concerns with three replies due to the length limit.
>
> - The presentation of FITS could be more comprehensive (more details in the questions). Furthermore, I would like to see more "basic" baselines (e.g., ARIMA, DeepAR for forecasting, and Random Cut Forest for Anomaly detection). To further substantiate the claims about performance (and to put them on a statistically sound foundation), a critical difference plot[1] may be a useful analysis tool. Lastly, the method employs several heuristics that need to be tuned in the real world, a drawback it shares with other methods.
>
>     A: *In line with previous studies like PatchTST, the effectiveness of statistical methods like ARIMA and RNN-based methods such as DeepAR and LSTM has been found to be inferior compared to Transformer-based methods. Hence, we did not include them as baselines in our study. Additionally, DeepAR is specially designed as a probabilistic forecasting model which may not be appropriate to compare with it. It's worth mentioning that RCF, a proprietary anomaly detection algorithm by Amazon, is not included in our analysis. In our manuscript, we present results for Isolated-Forecast, an ensemble and decision tree-based algorithm.*
>
>     *If we understand the ‘heuristic’ correctly,  FITS only have one hyper-parameter i.e., the cutoff frequency. It is easy to find the optimal value comparing to the large hyper-parameter space of other models.*
>
>     *We use the ARIMA implementation provided in the LTSF-Linear. However, as described in the repo, ARIMA runs extremely slow. As a result, we only report the MSE performance on ETTh1 and ETTm1.*
>
>     |Datasets|96|192|336|720|
>     |:---:|:---:|:---:|:---:|:---:|
>     |ETTh1|1.0642|1.1053|1.125|1.158|
>     |ETTm1|1.4870|1.7228|2.0984|3.3069|
>
>     *We have added the critical difference plot in the **Appendix I**. FITS outperforms all the baselines with statistical significance. The critical difference plot also showcases the robustness of FITS' performance across various datasets, reinforcing its reliability.*
>
> - Can you explain why x(t-τ) is a shift forward in time, rather than backward?
>
>     A: *When we talk about shifting forward in time, it means we are essentially repositioning the starting point of our time axis in the direction of increasing time values. Alternatively, you can think of it as sliding the entire function in the opposite direction, which is the negative part of the time axis. Consider a time series represented as F(t). Shifting forward by a duration of τ means that we are relocating the original reference point τ units ahead in time. This action results in a new representation of the function, denoted as F(t-τ). In simpler terms, it's like moving our viewpoint along the time axis to where τ becomes the new starting point.*
>
> - Can you add a more thorough caption to Fig. 2? As it stands, it is not comprehensible enough to fully understand your method. Is it just a single linear layer/what is the output dimension? What are the typical dimensions of ? Are (i)RIN and (i)rFFT performed in parallel or sequentially? Please elaborate more on how RIN works, either in the Figure or its caption.
>
>     A: *We have updated the caption for Figure 2 according to your advice.*
>
>     *Yes, as described in Section 3.2, the complex linear layer is just a single linear layer with complex-valued weights.*
>
>     *Suppose the input X dimension is (B, L, C). The $\tilde{X}$ should be the dimension of its rFFT which is (B, L//2+1, C). After a LPF, the $\tilde{X}$ is cut to the length of l according to the cutoff frequency. The input dimension of complex linear layer is (B, l, C) (Batch size, Length, Channel) and the output is (B, ηl, C) which extend the length according to the interpolation rate.*
>
>     *As mentioned in the paper, RIN stands for Reversable Instance Normalization. It is a simple normalization method that removes the bias and variance for each data instance before feeding into the model. After model processing, the variance and bias are added back to the result to reverse it back to the original data distribution. In most cases, RIN is used to eliminate the distribution shift so that models can reach a better result. However, FITS use RIN mainly to remove the bias of time series which can introduce a large 0 frequency component. Please check the paragraph under Figure 2 for detailed information.*
>
>     *RIN and FFT runs in sequential. As described, RIN is performed on the time domain. And we employ RIN to eliminate the bias in time series and use iRIN to add it back. Thus, RIN is performed ahead of the rFFT and iRIN in performed after the irFFT.*

---

> > ### Author Response · Authors · 2023-11-18
> >
> > - Have you experimented with more sophisticated layers than just linear?
> >
> >     A: *Thank you for raising this question. We appreciate your interest in exploring more sophisticated layers beyond linear ones in our study.*
> >
> >     *The frequency domain poses unique challenges due to its complex nature, and it is widely acknowledged that complex-valued neural networks are better suited for processing such data. We share your curiosity regarding the performance of complex-valued layers in our model. However, it is important to note that complex-valued neural networks are still in the early stages of development. Many commonly used components, including the Attention block, present mathematical obstacles when adapting them to the complex domain. Moreover, critical elements such as activation functions and loss functions are currently unavailable, making it difficult to construct large-scale neural networks.*
> >
> >     *Nonetheless, we recognize the potential of complex-valued neural networks in the frequency domain and plan to address this in our future work. We intend to investigate the use of complex-valued layers, such as a complex-valued frequency domain Transformer, to explore their impact on our model's performance.*
> >
> > - What is the back-casting task, and how is it used in training?
> >
> >     A: *As described in the fourth paragraph of Section 3.2. The back-casting is adding supervision on the reconstructed input. Since FITS outputs the extended time series segment which contains not only the forecasting horizon but also the reconstructed look-back window. By adding supervision on both of them, we were expecting a better performance.*
> >
> > - Can you motivate in 1-2 sentences why the output length needs to be controlled (Section 3.3)?
> >
> >     A: *According to the time series forecasting task definition, the model should make prediction on $x_{t, t+h}$ with given look-back window $x_{t-lbw, t}$. Generally, the forecasting model output length should be set as the length of target horizon. For FITS, we conduct the frequency interpolation on the input $x_{t-lbw, t}$ and generate the output as $x_{t-lbw, t+h}$. But anyway, the input and output length are decided according to the task setting (e.g.., input 720 output 96).*
> >
> > - You say the LPF discards components beyond the model's learning capabilities. What do you mean by this? The model's capabilities are independent of the signal, and high frequencies are not always noise or irrelevant. It seems domain-specific. Furthermore, you say high-frequency components such as trends are filtered out. First, trend signals can be low-frequency, and second, they are particularly important in the forecasting realm.
> >     1. *The primary objective of using a Low-Pass Filter (LPF) is to simplify the model by **making a trade-off**—specifically, by discarding high-frequency components that have minimal impact on the final prediction. As shown in Figure 3 of the original paper, when a significant portion of high-frequency components is removed, the overall shape of the time series remains largely unchanged. This suggests that if the model can effectively fit the primary frequencies, it can achieve good results with fewer parameters.*
> >     2. *FITS, being a compact linear model with very few parameters, struggles to handle data with complex patterns. High-frequency segments often contain information with extremely rapid temporal dynamics, such as noise. We believe that such information exceeds the learning capacity of FITS, which is why we employ the LPF to eliminate these high-frequency components and reduce the model's parameter count.*
> >     3. *As you rightly pointed out, the definition of "high-frequency" can vary across different datasets. Therefore, we conducted comprehensive ablation studies to demonstrate the impact of the cutoff frequency on the final prediction results across various datasets. This approach allows us to identify the primary frequency ranges that FITS can effectively learn, striking a balance between performance and memory usage.*
> >     4. *While trends are often considered low-frequency signals on a long time scale, time series analysis frequently operates on smaller segments of the data. In these cases, the long-term trend's influence on a particular segment can manifest as a bias. To preserve this bias and ensure it is not inadvertently removed by operations like LPF, we use a Reversable instance normalization (RIN). **Thanks for your suggestion, we have removed the description of ‘removing trend’ from the LPF part.***

---

> > > ### Author Response · Authors · 2023-11-18
> > >
> > > - The method is particularly interesting because of its parameter-effectiveness. Can you perform experiments on real-time inference time on simple hardware (not necessarily edge devices) to substantiate this (e.g., real-time anomaly detection)? What are the challenges regarding domain shifts and refitting, and how could they be addressed? This could substantially improve the focus and contribution of the paper.
> > >
> > >     A: *Indeed, we have taken steps to evaluate FITS on relatively simple hardware setups. We have reported the time consumption of FITS when executed on a single CPU core, a common scenario in edge devices. Furthermore, FITS can be directly deployed on ubuntu based single board computers (e.g. Raspberry Pi) with our original code. Similarly, FITS achieves inference time at around 3ms on Raspberry Pi.*
> > >
> > >     *Furthermore, we have also successfully deployed FITS on an ARM development board (specifically, the STM32F303K8T6, equipped with an ARM Cortex M4 core running at 72 MHz, 64 Kbytes of flash memory, and 16 Kbytes of SRAM). This board represents a computationally and memory-constrained device, making it a valuable test case.*
> > >
> > >     *Domain shift poses a common challenge in time series forecasting. Typically, it manifests as changes in bias and variance. To address this, we utilize RIN (instance normalization) during inference, effectively handling this problem. In cases of other shifts, such as frequency and waveform changes, we can employ off-the-shelf continual learning strategies to update FITS in real-time. FITS' computational and memory-efficient design allows for easy fine-tuning with minimal data on resource-constrained devices.*
> > >
> > > Hope our replies can address your concerns.

---

> > ### Comment · Reviewer_r1EX · 2023-11-21
> > **Thank you**
> >
> > Thank you for this response. Please double-check the interpretation of the critical difference plot. As your method is connected to the other methods, it is not statistically significantly better. However, your method is ranked on top, which remains a great signal. I would recommend reversing the x-order, though, so it is ranked first (not sixth). The current "last" ranking probably comes from the fact that the CD library expects the performance metric to be better when higher, which is not the case with MSE.

---

> ### Author Response · Authors · 2023-11-21
>
> We apologize for any confusion caused. As you correctly mentioned, we utilized a pre-existing repository for generating the critical difference (CD) plot. It is worth noting that the CD plot is commonly employed in classification tasks, where the evaluation metric is accuracy (higher is better). However, in our case, we evaluated a forecasting task based on mean squared error (MSE), where lower values indicate better performance. Consequently, being ranked sixth actually means that our model performs the best in terms of MSE.
>
> To create the CD plot, we originally only considered the MSE performance over a 336-forecasting horizon, and we have now updated the CD plot in accordance with all the evaluation settings. Kindly refer to the appendix for the updated version. We also attempted to reverse the order of the models in the plot. However, we encountered some issues with the plotting repository when activating the reversal. Consequently, we have retained the original order for now. We are actively working to rectify this issue and will update the diagram in the final version.
>
> The updated CD plot now demonstrates that our model, FITS, is no longer connected to the other methods, indicating statistical significance in its performance superiority.
>
> We apologize once again for any confusion caused and hope that our responses have adequately addressed your concerns.

---

> > ### Comment · Reviewer_r1EX · 2023-11-22
> > **Score increase**
> >
> > I thank the authors for their thorough rebuttal and increased my score by one as all the issues I had were resolved.

---

### Official Review · Reviewer_HyfR · 2023-10-31

**Soundness:** 3 good
**Presentation:** 4 excellent
**Contribution:** 3 good
**Rating:** 8
**Confidence:** 4

**Summary:**

The paper presents FITS, a lightweight model designed for time series analysis. In contrast to conventional models that work directly with raw time-domain data, FITS operates within the complex frequency domain. It utilizes a streamlined linear layer and an efficient low-pass filter, achieving state-of-the-art performance in forecasting and anomaly detection tasks with a mere 10k parameters. FITS offers a novel perspective on these tasks, viewing them as interpolation exercises within the frequency domain. This approach extends time series segments for forecasting and reconstructing downsampled data. FITS employs a complex-valued linear layer to master amplitude scaling and phase shift, facilitating efficient complex frequency domain interpolation. Its compact size and competitive performance render FITS an excellent choice for edge devices, unlocking a wide array of applications in time series analysis.

**Strengths:**

The paper stands out for its clear and well-structured writing, making it easy to grasp the presented ideas. Addressing a critical issue, the paper tackles the challenge of developing time series analysis models suitable for deployment on resource-constrained edge devices, ensuring optimal performance. Furthermore, the paper impressively conducts a comprehensive evaluation. It includes a comparison with various state-of-the-art methods, even though these methods vastly differ in size from the proposed approach. The paper also introduces an efficient method for selecting the cutoff frequency, explores the impact of different lookback window sizes through ablation studies, and offers detailed insights into the training process. Notably, the approach ingeniously combines simple components such as RevIN, LPF, and the Complex-valued Linear layer to create a robust architecture that can be applied to industrial-grade time series analysis tasks.

**Weaknesses:**

The method's limitation lies in its inability to generate probabilistic forecasts, a crucial requirement for numerous industrial applications. Moreover, the evaluations conducted on benchmark datasets may not accurately reflect the real-world scenarios of edge devices. These benchmark datasets, such as those related to traffic and weather, typically do not require processing on edge devices and can be handled in offline settings. It remains uncertain how well the model will perform when applied to data from edge devices, like healthcare devices or industrial sensors, which present different challenges and requirements.

**Questions:**

Did you try fitting a larger neural network for frequency interpolation task? It would be interesting to see if the performance of this architecture scales with the size.

---

> ### Author Response · Authors · 2023-11-18
>
> Thanks for your insightful questions.
>
> - The method's limitation lies in its inability to generate probabilistic forecasts, a crucial requirement for numerous industrial applications.
>
>     A: *We agree on the limitation that mentioned by the reviewer. As a compact model FITS is not designed to generate probabilistic forecasts. But it can be an interesting future work to generate probabilistic forecasts over frequency domain.*
>
> - Moreover, the evaluations conducted on benchmark datasets may not accurately reflect the real-world scenarios of edge devices. These benchmark datasets, such as those related to traffic and weather, typically do not require processing on edge devices and can be handled in offline settings. It remains uncertain how well the model will perform when applied to data from edge devices, like healthcare devices or industrial sensors, which present different challenges and requirements.
>
>     A: *We conduct evaluations on benchmark datasets primarily to ensure a fair and standardized comparison with other baseline models, demonstrating the performance of FITS in controlled conditions. We acknowledge that some of these benchmark datasets may not perfectly align with real-world edge device scenarios. However, it's important to note that our ETT dataset, sourced from sensors on an electrical transformer, closely matches our intended use case. Notably, FITS has demonstrated outstanding results on this dataset, reinforcing its suitability for relevant applications.*
>
>     *Furthermore, we are actively extending the applicability of FITS to various real-world tasks, including those involving healthcare devices. As part of this effort, we offer a script for generating an ONNX runtime, facilitating the deployment of FITS on embedded systems. We will further evaluate FITS on new datasets in our future works.*
>
> - Did you try fitting a larger neural network for frequency interpolation task? It would be interesting to see if the performance of this architecture scales with the size.
>
>     A: *Thank you for inquiring about this aspect. We share your curiosity regarding the performance of deep complex-valued neural networks in the context of frequency interpolation tasks. It's worth noting that the frequency domain inherently involves complex numbers, making complex-valued neural networks a natural choice for processing such data.*
>
>     *However, it's important to acknowledge that the development of complex-valued neural networks is still a work in progress. Critical components, including activation functions and loss functions tailored for complex numbers, are not widely available or well-established in the current landscape. This limitation poses challenges when attempting to construct large-scale complex-valued neural networks.*
>
>     *As part of our future research efforts, we are actively exploring strategies to overcome these obstacles and scale up complex-valued neural networks for applications in the frequency domain. This includes investigating complex-valued frequency domain Transformers as a potential avenue for advancement. Your question underscores the importance of this exploration, and we look forward to sharing our findings in this area as we make progress.*

---

> > ### Comment · Reviewer_HyfR · 2023-11-22
> >
> > Thanks to authors for answering my questions. After following the discussion with other reviewers, I'm increasing my score.

---

### Official Review · Reviewer_YAqa · 2023-11-01

**Soundness:** 3 good
**Presentation:** 3 good
**Contribution:** 3 good
**Rating:** 8
**Confidence:** 4

**Summary:**

This paper proposes an impressive compact model, named FITS, for time series tasks, including forecasting and anomaly detection. FITS achieves manipulation to time series through interpolation in the frequency domain. The whole framework is quite simple and has remarkably few parameters. FITS achieves competitive performance to SOTA baselines on both forecasting and anomaly detection with about 50 times fewer parameters. With such impressive performance, the proposed model would have a certain impact on the community.

**Strengths:**

1. The experiment result is surprisingly good considering the tiny footprint of the model. The standard deviation of the error is very small which may indicate that the model is very stable because of its simplicity.
2. Authors provide comprehensive ablation analysis to show light-weightness of the model and its superior performance across the hyper-parameters.
3. Authors use a synthetic dataset to show the key idea of FITS which is devide and conquer the different frequency. It also explained the effectiveness of the model on the AD task.

**Weaknesses:**

1. The result on anomaly detection is not remarkable. Especially on the SMAP and MSL dataset. Some more in depth analysis is needed to find out the reason.
2. Even though the FITS have far fewer parameters comparing to the DLinear, it still need more time to inference on the GPU (0.6 and 0.4 ms respectively). Does this mean the DLinear is still the best choice for the real-time application?

**Questions:**

1. How do author deploy the model on devices that do not support the complex computation?

---

> ### Author Response · Authors · 2023-11-18
>
> Thanks for your insightful questions.
>
> 1. The result on anomaly detection is not remarkable. Especially on the SMAP and MSL dataset. Some more in depth analysis is needed to find out the reason.
>
>     A: *As analyzed in the conclusion. These two datasets are majorly composed with binary data where time-domain modeling is preferable as the raw data format is sufficiently compact. The binary-valued data also have messy frequency presentation which is difficult for FITS to learn. We provide a visualization in the interpretability notebook in the anonymous repository to show such finding.*
>
> 2. Even though the FITS have far fewer parameters comparing to the DLinear, it still need more time to inference on the GPU (0.6 and 0.4 ms respectively). Does this mean the DLinear is still the best choice for the real-time application?
>
>     A: *The inference time of FITS is sub-ms level which is comparable to the communication overhead, the 0.2ms overhead does not make a dealbreaker to fail FITS to work as a real-time time series analysis model. Furthermore,* t*he inference time on GPU does not match the common case on the edge-device real-time application. The edge-device often only have single core computation unit which should resemble the inference time result on the single-CPU setting which FITs achieves better result. The inference time is also influenced by the cuda optimization. As far as we know, the complex-valued neural network in cuda and pytorch is still an experimental feature, which indicate that the optimization is still not as perfect as the normal linear layer. This may cause some performance downgrade. Furthermore, the edge devices often have very constrained memory resources such as the arm development board we use (STM32F303K8T6, which is with ARM Cortex M4 core at 72 MHz, 64 Kbytes of flash memory, 16 Kbytes of SRAM). The parameter of DLinear can not fit into a board with 64 KBytes of flash memory. As a result, FITS is still more suitable for the edge-device deployment.*
>
> 3. How do author deploy the model on devices that do not support the complex computation?
>
>     A: *We are aware of that many edge devices or even RTX4090 do not support complex computation. Thus, we provide a FITS-real model to simulate the complex-valued linear projection with normal real-valued linear layer as following:*
>
>     $$ y_{r}= W_r * x_r - W_i * x_i $$
>
>     and,
>
>     $$ y_{i}=W_r * x_i + W_i * x_r $$
>
>     *The complex-valued weight matrix is divided as two real-valued linear projection $W_i, W_r$.*
>
>     *The FITS-real model achieves the same result as the original FITS, and they have the same memory footprints.*

---

> > ### Comment · Reviewer_YAqa · 2023-11-21
> >
> > Thanks for the authors' rebuttal, all of my concerns have been properly addressed. I would like to keep my positive score.

---

### Public Comment · ~Jiacheng_You1 · 2023-11-29
**The number of parameters can be further reduced from a resampling perspective**

Dear authors:

I'd like to share my findings:
1. Since low pass filtering is equivalent to Fourier downsampling using [scipy.signal.resample](https://docs.scipy.org/doc/scipy/reference/generated/scipy.signal.resample.html) with `num = COF`, `FITS` is equivalent to perform `Linear` on downsampled time series, both input and label.
2. By performing `Linear` on downsampled time series directly, the number of parameters can be reduce from `dominance_freq * (dominance_freq * (seq_len + pred_len) / seq_len)` ($L'\times (L'+H')$, where $L',H'$ is downsampled or low pass filtered $L,H$) to `dominance_freq * (dominance_freq *  pred_len / seq_len)` ($L'\times H'$).

# 1

For simplicity, we omit `RevIN`.

Note that if we flatten $\mathbb{C}^{L/2}$ into $\mathbb{R}^L$, `rfft` can be viewed as a linear operator $D_L\in\mathbb{R}^{L\times L}$ from $\mathbb{R}^L$ to $\mathbb{R}^L$. At the same time, the Complex-valued Linear Layer become (let's omit `bias` first) a linear operator $W\in\mathbb{R}^{L'\times(L'+H')}$.

Note 1: Complex differentiable operators satisfy Cauchy-Riemann equations (complex linear operators are complex differentiable operators, or directly we have $f(ix)=if(x)=-\text{Im}[f(x)]+i\text{Re}[f(x)]\implies \text{Re}[f(ix)]=-\text{Im}[f(x)]\land \text{Im}[f(ix)]=Re[f(x)]$), thus the real form $\mathbb{R}^{L'\times(L'+H')}$ has "redundant" parameters comparing to the complex form $\mathbb{C}^{L'/2\times(L'+H')/2}$. To recover this behavior (and reduce parameters), a re-parameterization is needed. (BTW, I noticed that the number of parameters of `FITS` in Table 5 can be odd numbers. It seems that the authors count 1 complex number as 1 parameter, instead of 2.)

Note 1.1: All complex linear operators on $\mathbb{C}^{L/2}$ is a subset of all real linear operators on $\mathbb{C}^{L/2}$. For $\forall λ\in F, f(λx+y)=λf(x)+f(y)$, $F=\mathbb{C}$ gives "complex" linear, while $F=\mathbb{R}$ gives "real" linear. If the linear space is over $\mathbb{R}$, we can't define "complex" linear, since we might not have $λx \in X$.

Note 2: Flattening is an isomorphism between linear spaces $\mathbb{C}^{L/2}$ and $\mathbb{R}^L$.

Note 3: The proof does NOT depend on flattening. Because `rfft` and `irfft` are linear operators, `irfft(complex-linear(rfft(·)))` is a linear operator. Since both domain and co-domain of `irfft(complex-linear(rfft(·)))` are real vector spaces, this linear operator can be replaced by a real matrix.

Thus, `FITS` can be formulated as $D_{L+H}^{-1}\circ\text{pad}\circ W\circ\text{truncate}\circ D_L$.

Now we can insert identity $D_{L'}\circ D_{L'}^{-1}$ and $D_{L'+H'}\circ D_{L'+H'}^{-1}$.

We have $D_{L+H}^{-1}\circ\text{pad}\circ W\circ\text{truncate}\circ D_L = (D_{L+H}^{-1}\circ\text{pad}\circ D_{L'+H'})\circ(D_{L'+H'}^{-1}\circ W\circ D_{L'})\circ(D_{L'}^{-1}\circ\text{truncate}\circ D_L)$.

Note that $D_{L+H}^{-1}\circ\text{pad}\circ D_{L'+H'}$ is exactly a Fourier resampling from $L'+H'$ to $L+H$, and $D_{L'}^{-1}\circ\text{truncate}\circ D_L$ is a Fourier resampling from $L$ to $L'$.

It is obvious that $D_{L'+H'}^{-1}\circ W\circ D_{L'}$ is a linear operator $\tilde{W}\in\mathbb{R}^{L'\times(L'+H')}$ and we can easily add the `bias` back. From $D_{L'+H'}^{-1}(WD_{L'}z+b)=(D_{L'+H'}^{-1}WD_{L'})z+D_{L'+H'}b$, we have $\tilde{W}=D_{L'+H'}^{-1}WD_{L'}$ and $\tilde{b}=D_{L'+H'}^{-1}b$.

Now we have shown that `FITS` can be formulated as $\text{resample}^{L'+H'\to L+H}\circ \text{linear}\circ \text{resample}_{L\to L'}$.

We still need to show that the objectives are equivalent. We need to assume that MSE (i.e. 2-norm) is used. We denote the intermediate before `pad` in the original `FITS` formulation by $u = W\circ\text{truncate}\circ D_L (x)$.

`FITS` has an objective $||y-\hat{y}||^2 = ||y - D_{L+H}^{-1}\text{pad}(u)||^2=||D_{L+H}y - \text{pad}(u)||^2$, here we use 2-norm, and leverage the fact that Discrete Fourier Transform is 2-norm preserving.

`Linear` on downsampled time series has an objective $||D_{L'+H'}^{-1}\text{truncate}(D_{L+H}y) - D_{L'+H'}^{-1}u||^2=||\text{truncate}(D_{L+H}y) - u||^2$.

Obviously, the difference between these two objectives $||D_{L+H}y - \text{pad}(u)||^2 - ||\text{truncate}(D_{L+H}y) - u||^2$ is a constant with respect to $u$. Thus, the objectives are equivalent.

# 2
In the above section, we show that `FITS` is equivalent to perform `Linear` on downsampled time series, both input and label.

As a `Linear` on $L'\to(L'+H')$, it is trivial to show that the optimal solution of the $L'\to L'$ part is identity. Thus, we can completely drop the corresponding parameters.

$L'\to(L'+H')$ is introduced since `Typically, the forecasting horizon is shorter than the given look-back window, rendering direct
interpolation unsuitable.`. However, the resampling interpretation actually eliminate this concern.

---

### Meta-Review · Area_Chair_aBW9 · 2023-12-06

**Metareview:**

The paper proposes a parameter-efficient model architecture for time series anomaly detection and forecasting. The main advantage of the proposed model is to achieve comparable results with the state-of-the-art models with mere 10k parameters. The authors have addressed all the concerns raised by the reviewers. All the reviewers agree to accept the paper.

**Justification For Why Not Higher Score:**

The experiments are not really diverse.

**Justification For Why Not Lower Score:**

All the reviewers voted with high scores.

---

### Decision · Program_Chairs · 2024-01-16

Accept (spotlight)